# ExpertZIP: A Progressive Fusion Framework for Mixture-of-Experts Model Optimization through Huffman Tree Structures

## Abstract

Mixture-of-Experts (MoE) models have gained attention as a novel approach to developing large language models (LLMs), praised for their ability to enhance performance by utilizing multiple experts. However, while increasing the number of experts in these models can yield performance gains, it also introduces significant trade-offs, such as substantial memory overhead and increased inference time, limiting their scalability and practical deployment. In this work, we conduct a thorough analysis of expert utilization and identify inefficiency: many experts are underutilized, leading to suboptimal resource allocation with limited improvement. To address this issue, we propose ExpertZIP, a progressive framework for MoE models that leverages a Huffman tree-based expert fusion technique. This progressive approach systematically merges underutilized experts step by step, ensuring their essential contributions are maintained while drastically reducing memory usage and computational demands. Our approach yields a 17.23x reduction in model size and a 4.84x improvement in inference time, with only a 1.18% decrease in average accuracy compared to the original 64-expert Switch Transformer model. Moreover, it demonstrates a 6.47% increase in accuracy relative to models with an equivalent number of experts. These results demonstrate that our optimized framework provides performance on par with larger models, offering an efficient solution for resource-constrained and real-time applications.

## 1 Introduction

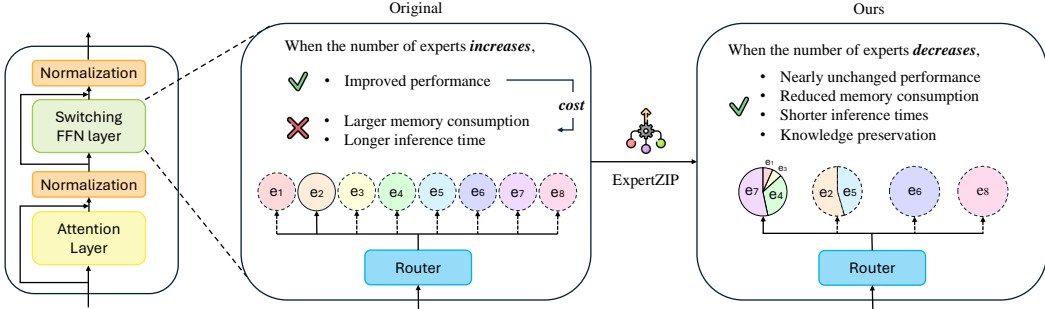

Figure 1: Comparison of original MoE architectures and ExpertZIP architectures. The original architecture suffers from higher memory consumption and longer inference times. In contrast, our approach achieves similar performance with reduced memory usage and shorter inference times.

The Mixture-of-Experts (MoE) model has become a prominent architecture in large language models (LLMs) Shazeer et al. (2017); Clark et al. (2022); Du et al. (2022); Zhou et al. (2022); Jiang et al. (2024), particularly for its ability to utilize different specialized experts to enhance performance across various natural language processing tasks. The core idea behind MoE is to train multiple specialized experts and use a gating network to dynamically select which experts should handle a given input, enabling efficient and targeted problem-solving. This selective routing enables MoE models

to scale effectively, significantly improving tasks such as machine translation, text generation, and question answering Lepikhin et al. (2021); Fedus et al. (2022).

Although adding more experts usually increases model performance, it also results in significantly higher memory usage and longer inference times (see Fig. 1 and 2). As the number of experts grows, the overall parameter count of the model expands substantially. Even though only a subset of experts is activated during inference, the parameters of all experts must be loaded and managed in memory, increasing both memory usage and access overhead. Furthermore, inference with different activated experts for different tokens further contributes to inference latency. These factors make deploying MoE models in real-world environments, particularly in

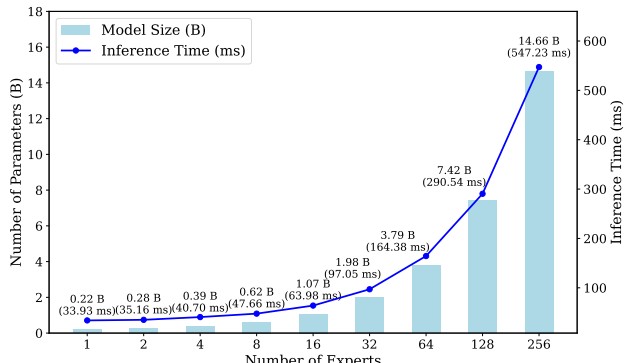

Figure 2: Effect of number of experts on model size and inference time in Switch Transformers.

resource-constrained settings, challenging. Balancing model performance with computational efficiency becomes crucial as models continue to scale. For example, deploying large MoE models in real-time applications, such as on mobile devices, can be both expensive and impractical.

Through our analysis of expert utilization within MoE models shown in Fig. 3, we observe that the usage of experts is often highly imbalanced. Many experts are underutilized, being rarely activated across different input samples. This imbalance is quantified by the selected frequency, representing the proportion of times an expert is chosen across the layers for a given dataset. The selected frequency for each expert in a layer indicates how often it is activated, with the sum of selected frequencies across all experts in a layer equal to 1. This imbalance suggests that the performance gains from adding more experts may not scale proportionally, as certain experts are not fully leveraged. Importantly, we also observe that even the less frequently used experts still contribute valuable information to the model's overall performance. This finding suggests that removing underutilized experts to reduce model complexity could result in losing important information, leading to a degradation in model accuracy.

To address these inefficiencies, we propose a novel approach called ExpertZIP (see Fig. 1) that enhances the efficiency of MoE models by merging underutilized experts rather than discarding them. Our method employs a weight combination strategy, utilizing Huffman tree Huffman (1952) to identify and combine the least significant experts. The Huffman tree, a well-known data structure used for data compression and encoding, suits our approach as it allows progressive merging of the least frequent experts and can preserve the most important one, perfectly aligning with our goal of balancing performance and efficiency. Additionally, since it is a tree structure, we can reduce the number of experts by stopping at a certain level. This approach ensures that the model retains the critical contributions of all experts while reducing redundancy and maintaining a compact architecture. Additionally, since we apply fusion to reduce the experts, we can preserve the original model's knowledge to achieve similar or surpassing performance compared to the original one.

This solution leads to a more efficient and deployable model architecture, making it suitable for real-world applications. By reducing the number of experts, we achieve significant savings in both memory usage and inference time. Our experiments demonstrate that the optimized model achieves performance comparable to or surpassing models with more experts. Specifically, in classification tasks, our method results in only a 1.18% drop in average accuracy, while in summarization tasks, the ROUGE-1 score Lin (2004) decreases by just 3.09%. Despite these minor performance reductions, the model achieves a 4.84x improvement in inference time and a 17.23x reduction in model size. As for comparing with the same number of experts after fusion, our approach can increase at most 6.47% and 7.74% on the classification and summarization task. These results validate the effectiveness of our approach, demonstrating its suitability for deployment in environments with limited resources but still preserving its performance, such as edge devices and real-time applications.

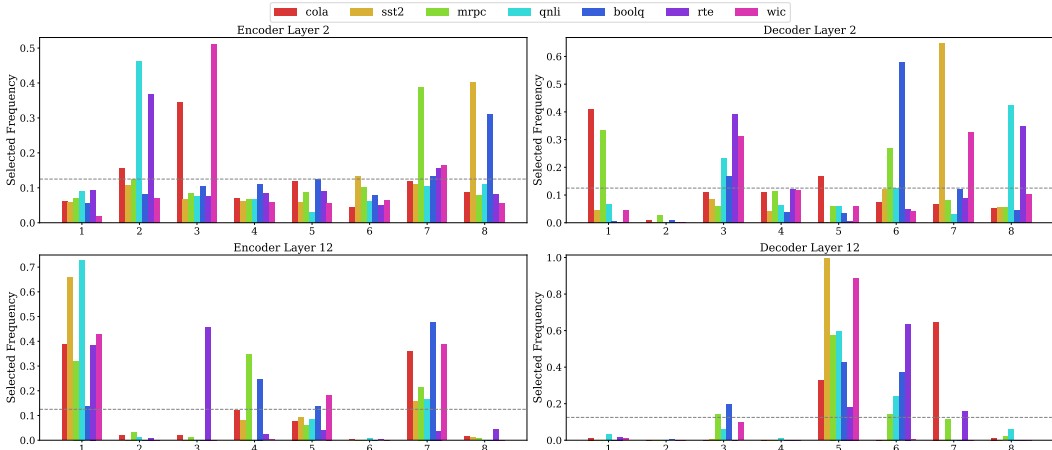

Figure 3: Expert selection distribution across different layers for the GLUE and SuperGLUE benchmarks is highly imbalanced with Switch Transformer. The data, aggregated from seven datasets within these benchmarks, shows that certain experts are selected frequently, while others are rarely, if ever, utilized. This indicates significant inefficiencies in the utilization of experts.

## 2 RELATED WORK

### 2.1 EXPERT MERGING IN MIXTURE-OF-EXPERTS MODELS

The Mixture-of-Experts (MoE) model has gained prominence in large language models (LLMs) for its scalable capacity Rajbhandari et al. (2022); Li et al. (2023a); Jiang et al. (2024); Cai et al. (2024). However, as complexity increases, efficient expert utilization and computational overhead become significant challenges. Recent methods have aimed to enhance efficiency and scalability while maintaining or improving performance through expert merging.

Key approaches include MEO He et al. (2023), which enables top-K expert selection without significantly increasing FLOPs. The Lory framework Zhong et al. (2024) optimizes auto-regressive models by using similarity-based data batching. ZIPIT Stoica et al. (2023) introduces a "zip" operation for merging disjoint tasks, aligning features across and within models for partial merging, effectively addressing feature compatibility issues. MC-SMoE Li et al. (2023b) leverages routing policies in sparse MoE to merge experts based on activation frequency, followed by compression to reduce memory and FLOPs. REPAIR Jordan et al. (2022) mitigates variance collapse in model merging by rescaling activations, improving interpolation performance and reducing accuracy barriers.

Most existing methods focus on model architecture, limiting the use of underutilized experts and failing to fully exploit their potential. Our approach addresses this by merging underutilized experts and adjusting the number of experts per layer, leading to more efficient capacity utilization and greater adaptability across configurations.

### 2.2 MODEL ENSEMBLE TECHNIQUES

Model ensemble techniques enhance robustness and performance in machine learning by leveraging model diversity, improving generalization, and mitigating overfitting. Key approaches include Bagging Breiman (1996); Khwaja et al. (2015); Błaszczyński & Stefanowski (2015), which reduces variance by training on data subsets, Boosting Freund et al. (1996); Waltner et al. (2019), which sequentially corrects errors, and Stacking Wolpert (1992); Low et al. (2019); Kang et al. (2020), which combines model outputs using a meta-learner.

Recent advancements have applied ensemble methods to deep learning, such as Deep Ensembles Lakshminarayanan et al. (2017); Buschjäger et al. (2020), which aggregate independently trained neural networks for improved performance and uncertainty estimation, and Snapshot Ensembles Huang et al. (2017); Zhang et al. (2020), which create ensembles within a single training run to minimize additional costs. Further innovations, like ME-TRPO Kurutach et al. (2018) for reinforce-

ment learning and SESoM PENG et al. (2023) for few-shot learning, highlight the growing use of ensemble techniques in various domains.

In this work, we integrate ensemble principles into MoE models by fusing underutilized experts. This approach retains diversity and robustness while significantly reducing computational overhead. Our method ensures that all experts' contributions are preserved, leading to a more efficient model that balances performance and complexity, particularly in resource-constrained environments.

## 3 BACKGROUND

### 3.1 MIXTURE OF EXPERTS

Mixture-of-Experts models offer an advanced approach to improving the efficiency and scalability of Transformer architectures. These models replace feed-forward network (FFN) layers with MoE layers, where only a subset of expert FFNs is activated for each input. In an MoE layer, $E$ experts are defined as $\text{FFN}(\cdot; \theta_1), \dots, \text{FFN}(\cdot; \theta_E)$, each mapping an input from $\mathbb{R}^d$ to $\mathbb{R}^d$. For a given input token $x$ with hidden state $h_x \in \mathbb{R}^d$, the routing function $R(h_x)$ selects the top $k$ experts, and the output $o_x \in \mathbb{R}^d$ is computed as: $o_x = \sum_{i \in \text{Top-}k(R(h_x))} e_i \cdot \text{FFN}(h_x; \theta_i)$, where $e_i = \text{Softmax}(R(h_x))_i$ represents the normalized routing score for the $i$-th expert.

### 3.2 HUFFMAN TREE

Huffman tree Huffman (1952) is a widely-used data structure for lossless data compression designed to merge the nodes based on frequencies. It starts by constructing a binary tree, where each leaf represents a symbol, and the path from the root defines the binary code for that symbol. Given a set of symbols with frequencies, the algorithm iteratively merges the two least frequent symbols, assigns binary digits to each branch, and recalculates combined frequencies until only one node remains. In the context of our designed MoE models, each expert corresponds to a symbol, with activation frequency analogous to symbol frequency.

## 4 METHODOLOGY

We aim to enhance the efficiency of MoE models by strategically reducing the number of experts while preserving or improving overall performance. Our approach is designed to facilitate the fusion of any number of experts, ensuring the retention of essential knowledge throughout the process. The methodology is structured into expert selection via the Huffman tree, expert fusion through a progressive weighted sum approach, and fine-tuning of the fused model. This process ensures that the contributions of each expert are preserved during fusion, allowing for flexible reduction without compromising the model's integrity.

### 4.1 EXPERT SELECTION STRATEGY USING HUFFMAN TREE

The first phase of our approach involves selecting which experts to fuse based on their utilization frequency within the MoE model. Our analysis has revealed that there is often a significant imbalance in expert utilization, where specific experts are underutilized. This underutilization suggests that these experts contribute less to the overall performance, making them prime candidates for fusing. To systematically identify and select these underutilized experts, we employ the Huffman tree, a well-known data structure for Huffman coding. Huffman tree is particularly suited for this task because it optimally fuses elements with the lowest frequencies, which aligns with our objective of reducing the number of underutilized experts and supporting the fusion of any desired number of experts by halting at a specified stage.

For each MoE layer, we calculate the selected frequency of the experts based on the training data. The distribution is denoted as $F^l = \{f_1^l, f_2^l, \dots, f_N^l\}$, where $f_i^l$ represents the selected frequency of the $i$-th expert at the $l$-th layer. The frequency $f_i^l$ is computed as the proportion of input tokens in the dataset for which expert $e_i$ is among the top-$k$ experts selected by the routing function $R(h_x^l)$, where $h_x^l$ is the hidden state of input token $x$ at the $l$-th layer, and $D$ represents the set of input tokens in the training data. The selected frequency $f_i^l$ is formally calculated as:

$$f_i^l = \frac{1}{k|D|} \sum_{x \in D} \mathbf{1}_{\{e_i \in \text{Top-}k(R(h_x^l))\}}, \tag{1}$$

where $\mathbf{1}_{\{.\}}$ represents the indicator function. In our study, we set $k = 1$, meaning that for each input token, only the top-1 expert is selected. To control the reduction rate of experts, we define a variable called *speed*, which dictates the rate at which experts are fused and reduced. For example, if the original number of experts is 16 and the speed is set to 2, the experts are reduced to 8 before fine-tuning. The *speed* variable allows flexibility in choosing the granularity of reduction and ensures that fine-tuning is performed at each step to maintain optimal performance.

The Huffman tree process then fuses the two experts with the lowest frequencies iteratively, forming a new expert whose frequency is the sum of the frequencies of the original two experts. This process is repeated until the desired number of experts, $N_e$, is achieved. In practice, we use a min heap to efficiently extract the node with the minimum frequency for each step. To illustrate the fusing process, we provide an example in Fig. 4 where eight experts are gradually merged into one expert. We also provide the detailed steps of our Huffman tree-based expert fusing process in Alg. 1.

### 4.2 WEIGHTED SUM FUSING OF EXPERTS

The second phase of our methodology involves fusing the experts identified in the previous phase. Once the Huffman tree process selects the underutilized experts, we fuse these experts by computing a weighted sum of their parameters, specifically designed to preserve knowledge. Formally, for the $l$-th MoE layer, let $E^l = \{e_1, e_2, \cdots, e_N\}$ represent the original set of experts, and $F^l$ denotes their corresponding selected frequencies. The set of fusing indices provided by the Huffman tree is denoted by $M_f$. The fusing process begins by initializing each new expert $e_n'$ in the new set $E^{l'}$ with a zero weight. For each original expert $e_m$ identified for fusion, its parameters $W_{e_m}$ is scaled by its selected frequency $f_m^l$ and added to a new weight $W_{e_n'}$ for expert $e_n'$. After aggregating all contributions, the weight of $e_n'$ is normalized by the total frequency selected by the merged experts. This normalization step ensures that the overall magnitude of the weights remains consistent, thereby preserving model stability and performance. The weight $W_{e_n'}$ for a new expert $e_n'$ is computed as:

$$W_{e_n'} = \frac{\sum_{m \in M_f[n]} W_{e_m} \times f_m^l}{\sum_{m \in M_f[n]} f_m^l}, \tag{2}$$

where $M_f[n]$ represents the merging set that forms the new expert $e_n'$. Using a weighted sum based on selected frequencies, we ensure that the most important characteristics of the original experts are preserved during the fusion process, preventing loss of critical knowledge. Alg. 2 outlines the detailed steps involved in this expert fusing process. This approach not only reduces redundancy but also maintains the model's overall performance. Once the fusion process is complete, we further fine-tune the model on downstream tasks to optimize its performance and compensate for potential loss of accuracy during the fusion phase. This fine-tuning step ensures the fused model reaches its maximum potential, restoring performance drop and maintaining robustness across tasks.

## 5 EXPERIMENTS

### 5.1 SETTINGS

Our experiments utilize the Switch Transformers Fedus et al. (2022). We select the model with 8, 16, 32, and 64 experts. Models with a larger number of experts are excluded due to hardware limitations. The experiments are conducted on an NVIDIA A100 GPU with 80GB of memory. Detailed hyperparameters of the experiments can be found in Appx. D. Unless otherwise specified, we reduce the number of experts by setting the *speed* to 2.

### 5.2 DATASET

To evaluate our proposed method, we conduct experiments on both classification and summarization tasks using datasets from GLUE, SuperGLUE, and CNN/Daily Mail (CNNDM). For the classification tasks, we select CoLA Warstadt et al. (2019), SST-2 Socher et al. (2013), MRPC Dolan & Brockett (2005), and QNLI Rajpurkar et al. (2016) from the GLUE benchmark, which tests various

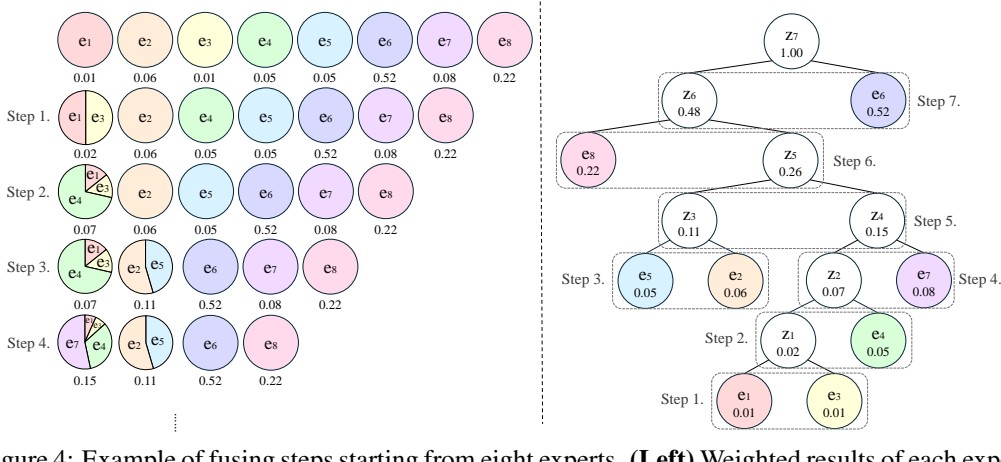

Figure 4: Example of fusing steps starting from eight experts. **(Left)** Weighted results of each expert. **(Right)** Alternative illustration with Huffman tree.

Table 1: Performance results after applying ExpertZIP, with experts reduced *speed*=2.

| Original Size | Classification (Accuracy ↑) | | | | Summarization (ROUGE-1 ↑) | | | |
|---|---|---|---|---|---|---|---|---|
| | 64 | 32 | 16 | 8 | 64 | 32 | 16 | 8 |
| 64 | 78.62 | **78.30** (+1.28%) | **77.72** (+2.80%) | **77.65** (+6.47%) | 36.19 | **35.88** (+0.06%) | **36.05** (+4.98%) | **35.35** (+7.74%) |
| 32 | - | 77.31 | 76.89 (+1.71%) | 76.61 (+5.05%) | - | 35.86 | 34.70 (+1.05%) | 34.26 (+4.42%) |
| 16 | - | - | 75.60 | 75.08 (+2.94%) | - | - | 34.34 | 33.40 (+1.80%) |
| 8 | - | - | - | 72.93 | - | - | - | 32.81 |

aspects of language understanding such as grammatical acceptability, sentiment analysis, sentence equivalence, and question answering. Additionally, we include BoolQ Clark et al. (2019), RTE Dagan et al. (2005), and WiC Pilehvar & Camacho-Collados (2019) from the SuperGLUE benchmark, focusing on tasks like question answering, textual entailment, and word sense disambiguation. For the summarization task, we utilize the CNN/Daily Mail (CNNDM) dataset (Nallapati et al., 2016), which pairs news articles with human-written summaries, offering a rigorous test of a model's ability to generate concise and coherent summaries from longer texts.

## 5.3 EVALUATION METRICS

To evaluate the performance of our proposed method, we use different metrics depending on the task. For classification tasks, we use accuracy, the ratio of correctly predicted labels to the total number of samples. A higher accuracy score indicates better classification performance, as more predictions align with the true labels. For summarization tasks, we use the ROUGE-1 Lin (2004), which measures the overlap of unigrams (single words) between the generated summary and the reference summary (ROUGE-2 and ROUGE-L are reported in Appx. I). The higher the ROUGE-1 is, the more overlap exists, reflecting better summarization quality in capturing important information from the reference text. The reported results are all re-implemented and fine-tuned by ourselves.

## 5.4 RESULTS

**Knowledge Preservation.** In Tab. 1, we present the outcomes of applying ExpertZIP for classification and summarization tasks, demonstrating that the knowledge from the large experts can be preserved. Starting with 16, 32, and 64 experts, the number of experts is halved after each fusion step. When reducing the number of experts from A to B (A > B), the performance on each task is better than the original model with B experts. For instance, classification tasks see performance

Table 2: Performance results of ExpertZIP and ExpertZIP* starting from 64 experts on classification tasks. ExpertZIP halves the number of experts at each step, while ExpertZIP* gradually reduces by halving until 16 experts, then one at a time.

| Method | # of Experts | CoLA | SST2 | MRPC | QNLI | BoolQ | RTE | WiC | Average ↑ |
|---|---|---|---|---|---|---|---|---|---|
| SwitchT. | 64 | 82.65 | 94.72 | 84.31 | 91.60 | 71.71 | 68.59 | 56.74 | 78.62 |
| ExpertZIP | 32 | 81.50 | 93.81 | 85.05 | 90.76 | 72.26 | 68.95 | 55.80 | 78.30 (-0.41%) |
| | 16 | 78.24 | 92.32 | 84.56 | 90.08 | 72.72 | 69.68 | 56.43 | 77.72 (-1.14%) |
| | 8 | 76.70 | 91.40 | 83.58 | 89.97 | 72.66 | 70.40 | 55.64 | 77.19 (-1.82%) |
| | 4 | 75.55 | 91.17 | 84.31 | 89.68 | 72.69 | 70.40 | 56.27 | 77.15 (-1.87%) |
| | 2 | 75.84 | 91.28 | 85.29 | 89.84 | 72.97 | 68.95 | 55.49 | 77.09 (-1.95%) |
| | 1 | 75.26 | 91.28 | 84.07 | 90.01 | 73.39 | 68.23 | 56.89 | 77.02 (-2.04%) |
| ExpertZIP* | 8 | 77.85 | 92.55 | 84.31 | 90.10 | 73.61 | 69.31 | 55.80 | **77.65 (-1.23%)** |
| | 4 | 77.28 | 92.20 | 84.07 | 90.06 | 73.70 | 69.68 | 57.05 | **77.72 (-1.14%)** |
| | 2 | 76.80 | 92.09 | 85.05 | 89.93 | 73.91 | 69.31 | 57.21 | **77.76 (-1.09%)** |
| | 1 | 77.09 | 92.09 | 84.80 | 90.30 | 73.52 | 68.95 | 57.05 | **77.69 (-1.18%)** |

Table 3: (**Left**) Performance results of ExpertZIP on the summarization task, starting with 64 experts with *speed* = 2. (**Right**) Comparison of model size and inference time between the original 64-expert model and progressively smaller expert configurations.

| Method | # of Experts | ROUGE-1 ↑ | # of Experts | Model Size (B) ↓ | Time (ms) ↓ |
|---|---|---|---|---|---|
| SwitchT. | 64 | 36.19 | 64 | 3.79 | 164.38 |
| ExpertZIP | 32 | 35.88 (-0.86%) | 32 | 1.98 (1.91x) | 97.05 (1.69x) |
| | 16 | 36.05 (-0.39%) | 16 | 1.07 (3.54x) | 63.98 (2.57x) |
| | 8 | 35.35 (-2.32%) | 8 | 0.62 (6.11x) | 47.66 (3.45x) |
| | 4 | 35.28 (-2.51%) | 4 | 0.39 (9.72x) | 40.70 (4.04x) |
| | 2 | 35.56 (-1.74%) | 2 | 0.28 (13.54x) | 35.16 (4.68x) |
| | 1 | 35.07 (-3.09%) | 1 | **0.22 (17.23x)** | **33.93 (4.84x)** |

gains of up to 6.47% (64 → 8) compared to the Switch Transformers originally have only 8 experts, while summarization tasks experience up to a 7.74% improvement (64 → 8) in ROUGE-1 scores. This demonstrates that ExpertZIP effectively preserves the model's knowledge, allowing for the reduction of expert numbers without sacrificing crucial information or performance.

**Unchanged Performance with Smaller Size and Faster Speed.** Tab. 2 further illustrates the performance of ExpertZIP and ExpertZIP* on classification tasks. ExpertZIP reduces the number of experts by half at each step, while ExpertZIP* employs a slower reduction strategy, halving until 16 experts and then reducing one at a time. Both approaches show minimal performance loss compared to the Switch Transformers 64-expert configuration. ExpertZIP* demonstrates slightly better preservation of accuracy, with average accuracy dropping by only 1.18%, while ExpertZIP results in a decrease of 2.04%. This highlights the effectiveness of these strategies in maintaining strong performance while reducing the model's complexity. Tab. 3 (left) focuses on the summarization task, showing that even when the number of experts is reduced to as low as one, the ROUGE-1 score remains competitive, with only a 3.09% drop compared to the switch transformers 64-expert model. Tab. 3 (right) highlights the significant reductions in model size and inference time. By reducing the number of experts, the model size shrinks by up to 17.23x and inference time decreases by 4.84x when reduced to a single expert. These results demonstrate the efficiency of ExpertZIP in drastically improving resource usage without sacrificing much performance.

**Comparison with Other Methods.** Tab. 4 compares the performance and model size of our proposed ExpertZIP approach with existing pruning, quantization, and merging methods on GLUE and SuperGLUE. The results show that ExpertZIP achieves competitive performance with significantly reduced model size, highlighting its efficiency in preserving knowledge while minimizing resource requirements. Notably, the enhanced variant, ExpertZIP*, consistently outperforms other methods, achieving the highest average score with the lowest performance degradation (-1.18%) compared to the original 64-expert Switch Transformer model. These results demonstrate the effectiveness of ExpertZIP in maintaining model robustness while optimizing computational efficiency.

Table 4: Performance comparison between ExpertZIP (and its enhanced variant ExpertZIP*) and other compression methods across GLUE and SuperGLUE benchmarks.

| Category | Method | Model Size | CoLA | SST2 | MRPC | QNLI | BoolQ | RTE | WiC | Average ↑ |
|----------|--------|-----------|------|------|------|------|-------|-----|-----|-----------|
| | SwitchT. (64 expert) | 3.79B | 82.65 | 94.72 | 84.31 | 91.60 | 71.71 | 68.59 | 56.74 | 78.62 |
| Pruning / Quntization | Task-Specific Chen et al. (2022) | 0.22B | 70.25 | 86.43 | 75.76 | 80.24 | 67.24 | 65.43 | **57.21** | 71.79 (-8.69%) |
| | PS-MoE Lu et al. (2024) | 0.22B | 72.14 | 87.18 | 74.38 | 83.25 | 70.75 | 64.97 | 56.43 | 72.73 (-7.49%) |
| | UV-MoE He et al. (2024) | 0.22B | 71.25 | 88.47 | 80.23 | 79.67 | 71.71 | 65.43 | 56.89 | 73.38 (-6.67%) |
| Merging | REPAIR Jordan et al. (2022) | 0.52B | 74.02 | 90.21 | 83.27 | 86.75 | 73.39 | 67.23 | 58.42 | 76.18 (-3.10%) |
| | ZipIt Stoica et al. (2023) | 0.52B | 73.98 | 91.78 | 82.58 | 87.90 | 72.14 | 66.18 | 57.05 | 75.94 (-3.41%) |
| | M-SMoE (1 expert) Li et al. (2023b) | 0.52B | 75.18 | **92.43** | 83.53 | 88.94 | **74.29** | 68.59 | 56.43 | 77.06 (-1.98%) |
| Ours | ExpertZIP (1 expert) | 0.22B | 75.26 | 91.28 | 84.07 | 90.01 | 73.39 | 68.23 | 56.89 | 77.02 (-2.04%) |
| | ExpertZIP* (1 expert) | 0.22B | **77.09** | 92.09 | **84.80** | **90.30** | 73.52 | **68.95** | 57.05 | **77.69 (-1.18%)** |

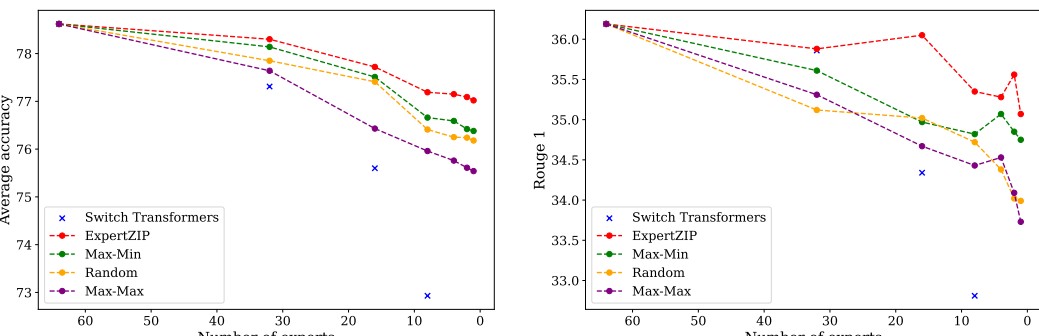

Figure 5: Comparison of different expert selection strategies.

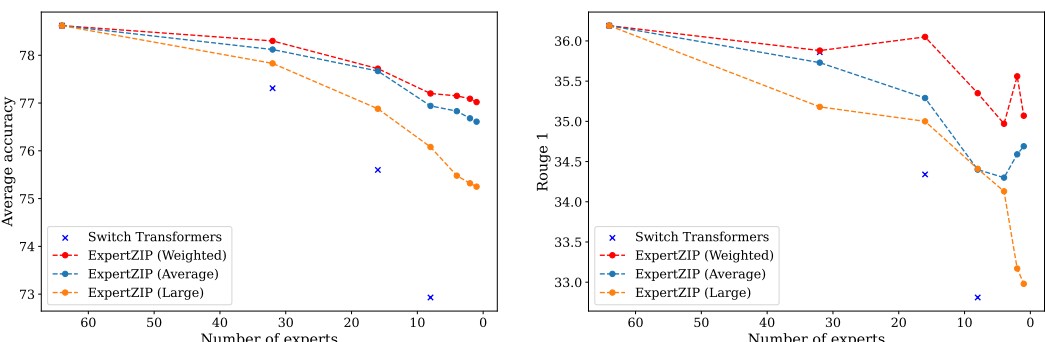

Figure 6: Comparison of different weight fusion strategies.

## 5.5 ABLATION STUDY

**Expert Selection.** We compare four methods based on performance to identify how more balanced expert combinations lead to better results. The Max-Max strategy, which pairs experts with the highest selected frequencies, creates a more imbalanced expert selection. The random strategy, pairing experts randomly, also fails to produce improvements as the lack of structure in expert selection leads to unpredictability. In contrast, the Max-Min strategy, which pairs experts with the highest and low-

est frequencies, achieves a more balanced contribution, resulting in better performance than Max-Max and random strategies. Our proposed ExpertZIP method, which ensures both flexibility and a balanced distribution of expert contributions, achieves the best performance because it not only promotes balance but also leverages the Huffman tree structure. This approach minimizes the impact on high-frequency experts, focusing the merging process on lower-frequency experts, thereby preserving the performance of the most important experts (see Fig. 5).

**Weight Fusion.** We conduct a comparative analysis involving three fusion methods to show that our fusion approach is the most effective. Fig. 6 presents the results of our proposed method alongside two alternative strategies. The ExpertZIP (Large) method retains only the expert with the highest selected frequency, the ExpertZIP (Average) method assigns equal weights to all selected experts, and the ExpertZIP (Weighted) method employs a weighted sum based on the relative importance of each expert. The experimental outcomes highlight that the ExpertZIP (Weighted) approach yields the best performance, as it effectively captures the differential contributions of each expert, leading to superior overall results. Notably, the inferior performance of the Large method, which removes less frequently selected experts, reinforces the observation made in our introduction that even underutilized experts can contribute valuable information to the model's overall performance. This supports the idea that removing these experts outright may result in losing critical information, leading to performance degradation. Thus, our findings highlight the importance of retaining and fusing expert contributions, even from less frequently activated experts, to maintain model accuracy.

**Fusion Speed.** We evaluated the effect of varying speed approaches on model performance. We tested five different speeds, including a constant rate (decreasing the number of experts by 1 in each step) and speeds of 2, 4, 8, and 16, where each speed represents dividing the number of experts by the corresponding factor at each step. The experimental results, shown in Fig. 7, indicate that slower reduction speeds yield better performance. Specifically, the constant speed and lower division factors preserve model accuracy more effectively, while faster speeds lead to greater performance degradation.

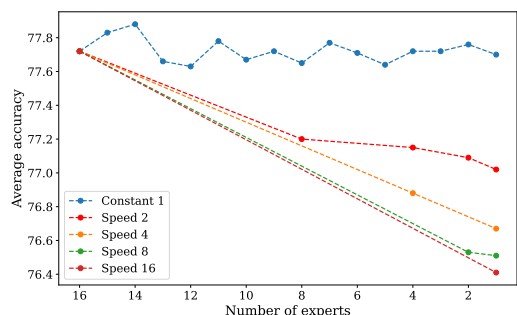

Figure 7: Comparison across different speeds.

## 6 DISCUSSION

In this section, we explore additional architectures to evaluate the robustness and adaptability of ExpertZIP. Specifically, we integrate ExpertZIP into the Mixtral 8x7b model Jiang et al. (2024) to address a limitation in Switch Transformers Fedus et al. (2022). While Switch Transformers is powerful, it requires fine-tuning for each downstream task. Each expert in Switch Transformers is tailored for a single task, reducing adaptability across tasks. In contrast, Mixtral 8x7b leverages 8 experts to handle various tasks simultaneously, enhancing generalization through task diversity. This makes Mixtral 8x7b an ideal candidate to test whether ExpertZIP can benefit without compromising the model's generalization capabilities. Therefore, we conduct additional experiments to evaluate ExpertZIP's effectiveness in this more versatile setting.

Given this key difference, we aim to investigate whether ExpertZIP, shown to optimize expert utilization in other MoE models, remains effective in a scenario where generalization plays a more prominent role, as in the Mixtral 8x7b model. By incorporating ExpertZIP into Mixtral 8x7b, we aim to determine if it could still provide the same benefits and performance maintenance with expert fusing without compromising the model's ability to generalize across various downstream tasks. Due to limitations in hardware resources, we opt to experiment with a speed 4 configuration, where the number of experts is progressively reduced from 8 to 2 experts. We then fine-tune the model for 1 epoch on the RedPajama-1B dataset, a large-scale dataset designed for pretraining language models Computer (2023). This experiment allows us to test ExpertZIP's ability to optimize large MoE models while retaining essential performance metrics.

Additionally, we investigate the effects of fine-tuning on varying amounts of data, specifically 1%, 5%, and 10% of the dataset, to evaluate the performance trade-offs at different data scales. To comprehensively assess the model's generalization capabilities, we expand our evaluation beyond traditional benchmarks like GLUE and SuperGLUE Wang et al. (2019b;a) by incorporating a wider array of challenging tasks such as MMLU Hendrycks et al. (2020), HellaSwag Zellers et al. (2019), PIQA Bisk et al. (2020), ARC (Easy and Challenge) Clark et al. (2018), MathQA Amini et al. (2019), and Winogrande Sakaguchi et al. (2021). These benchmarks provide a diverse range of reasoning, com-

Table 5: Performance comparison between the original Mixtral 8x7b model and the ExpertZIP model across various GLUE and SuperGLUE tasks on the RedPajama-1B dataset.

| Method | # of Experts | % of Dataset | CoLA | SST2 | MRPC | QNLI | BoolQ | RTE | WiC | Average ↑ |
|---|---|---|---|---|---|---|---|---|---|---|
| Mixtral 8x7b | 8 | | 66.25 | 85.55 | 73.04 | 58.39 | 85.38 | 70.04 | 60.19 | 71.26 |
| ExpertZIP | 2 | 1% | 65.58 | 77.06 | 69.36 | 55.23 | 76.64 | 62.12 | 52.83 | 65.55 (-8.01%) |
| | | 5% | 64.71 | 80.80 | 68.42 | 53.98 | 80.09 | 65.57 | 55.67 | 67.03 (-5.94%) |
| | | 10% | 65.03 | 83.65 | 67.99 | 57.04 | 81.35 | 66.55 | 56.98 | 68.43 (-3.97%) |

Table 6: Performance comparison between the original Mixtral 8x7b model and the ExpertZIP model across diverse reasoning and commonsense tasks on the RedPajama-1B dataset.

| Method | # of Experts | % of Dataset | MMLU | HellaS | PIQA | Arc-e | Arc-c | MathQA | WinoG | Average ↑ |
|---|---|---|---|---|---|---|---|---|---|---|
| Mixtral 8x7b | 8 | | 68.81 | 67.65 | 83.57 | 76.81 | 56.23 | 36.98 | 77.43 | 66.45 |
| ExpertZIP | 2 | 1% | 58.47 | 66.92 | 76.61 | 68.61 | 48.16 | 30.14 | 69.35 | 59.75 (-10.08%) |
| | | 5% | 63.57 | 71.02 | 78.08 | 72.61 | 52.53 | 32.17 | 70.38 | 62.91 (-5.33%) |
| | | 10% | 66.13 | 70.85 | 79.55 | 71.64 | 55.27 | 32.44 | 73.67 | 64.22 (-3.36%) |

monsense, and multiple-choice question tasks, which test the model's ability to generalize beyond its training data.

After applying ExpertZIP to reduce the number of experts and fine-tune on these benchmarks, the performance on 10% of the RedPajama-1B dataset drop by 3.97% on GLUE and SuperGLUE tasks (see Tab. 5), and 3.36% on reasoning and commonsense tasks (see Tab. 6). These minimal perfor-

Table 7: Comparison of size and inference time.

| # of Experts | Model Size (B) ↓ | Time (ms) ↓ |
|---|---|---|
| 8 | 46.70 | 756.07 |
| 2 | **12.88 (3.63x)** | **650.93 (1.21x)** |

mance decreases illustrate that ExpertZIP effectively reduces the model size without significant accuracy loss, even when applied to larger and more complex models. Additionally, the model size is reduced by 3.63x, and inference time is shortened by 1.21x (see Tab. 7), demonstrating the efficiency gains provided by ExpertZIP. Furthermore, this result shows that ExpertZIP maintains the model's generalization ability across different datasets and diverse task domains, confirming that the technique is scalable and adaptable to a broader range of real-world scenarios.

# 7 CONCLUSION AND FUTURE WORK

In this paper, we present ExpertZIP, a novel framework for optimizing Mixture-of-Experts models by fusing underutilized experts through a Huffman tree-based expert fusion technique. Our approach addresses the inefficiencies caused by imbalanced expert utilization, significantly reducing model size and inference time while maintaining near-equivalent performance. Specifically, we demonstrated that our method, applied to the Switch Transformer model, achieves a 17.23x reduction in model size and a 4.84x improvement in inference time, with only a 1.18% drop in accuracy for classification tasks and a 3.09% drop in ROUGE-1 score for summarization tasks. Furthermore, when compared with Switch Transformer models having the same number of experts after fusion, our approach shows improvements of up to 6.47% on classification tasks and 7.74% on summarization tasks. Additionally, we successfully test ExpertZIP on the Mixtral 8x7b model, designed to handle multiple tasks simultaneously and enhance generalization. Our results confirm that ExpertZIP remains effective even on this more generalized architecture, demonstrating its scalability and adaptability. In our future planning, we aim to investigate strategies for dynamically adjusting the number of experts at different layers. This could lead to even greater improvements in model efficiency and performance, expanding the applicability of MoE models to a broader range of real-world scenarios.

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

## A  ALGORITHM

In this section, we provide the detailed algorithms that form the core of our approach for compressing Mixture of Experts (MoE) models. Algorithm 1 outlines the Huffman tree-based method for selecting and fusing underutilized experts, leveraging their utilization frequencies to identify candidates for merging systematically. This process ensures that the most redundant experts are prioritized for fusion, guided by the principles of Huffman coding.

Algorithm 2 presents the weighted sum fusion process, where the parameters of the selected experts are aggregated based on their utilization frequencies. This technique ensures that critical knowledge from the original experts is preserved, while maintaining model stability and performance. These algorithms work in tandem to achieve significant compression in the number of experts, balancing model efficiency and effectiveness.

---

**Algorithm 1** Huffman Fusing

---

**Input:** Selected frequency $F = \{f_1, f_2, \cdots, f_N\}$, number of required expert $N_e$
**Output:** List of fusing indices $M_f$
1: $heap \leftarrow \text{heapify}([f_{\{1\}}, f_{\{2\}}, \cdots, f_{\{N\}}])$           ▷ *Creating a data heap*
2: $M_f \leftarrow []$
3: **while** size of $heap > N_e$ **do**
4:     $f_x \leftarrow$ Extract minimum element from $heap$
5:     $f_y \leftarrow$ Extract minimum element from $heap$
6:     $f_{x \cup y} \leftarrow f_x + f_y$; Insert $f_{x \cup y}$ into $heap$     ▷ *Fusing two experts and insert back to heap*
7: **end while**
8: **while** size of $heap > 0$ **do**
9:     $f_{\mathbf{K}} \leftarrow$ Extract minimum element from $heap$
10:     Append $\mathbf{K}$ into $M_f$
11: **end while**
12: **return** $M_f$

---

**Algorithm 2** Fusing Expert for $l$-th MoE Layer

---

**Input:** Original expert for $l$-th MoE layer $E^l = \{e_1, e_2, \cdots, e_N\}$, selected frequency for $l$-th MoE layer $F^l = \{f_1^l, f_2^l, \ldots, f_N^l\}$, and number of required expert $N_e$
**Output:** New expert for $l$-th MoE layer $E^{l'}$
1: $E^{l'} \leftarrow \{\}$
2: $M_f \leftarrow \texttt{Huffman\_Fusing}(F^l, N_e)$     ▷ *Fuse experts based on their frequencies*
3: **for** each $n$ in $(1 \cdots N_e)$ **do**
4:     $e_n' \leftarrow$ Initialize a new MoE layer with weight of zeros
5:     $count \leftarrow 0$                  ▷ *Accumulate the total selected frequency*
6:     **for** each original expert $k$ in $M_f[n]$ **do**
7:        $e_n' \leftarrow e_n' + e_m \times f_m^l$
8:        $count \leftarrow count + f_m^l$
9:     **end for**
10:     $E^{l'} \leftarrow E^{l'} \cup (e_n'/count)$
11: **end for**
12: **return** $E^{l'}$

---

## B  VISUALIZATION OF EXPERT IMBALANCE

In this section, we present the visualization of expert selection imbalance across different layers in the MoE model. Fig. 8, 9, and 10 illustrate the selected frequency distribution of experts for models with 8, 16, and 32 experts, respectively, across various encoder and decoder layers. These visualizations highlight how specific experts are selected more frequently than others, indicating an imbalance in the utilization of the experts in each layer.

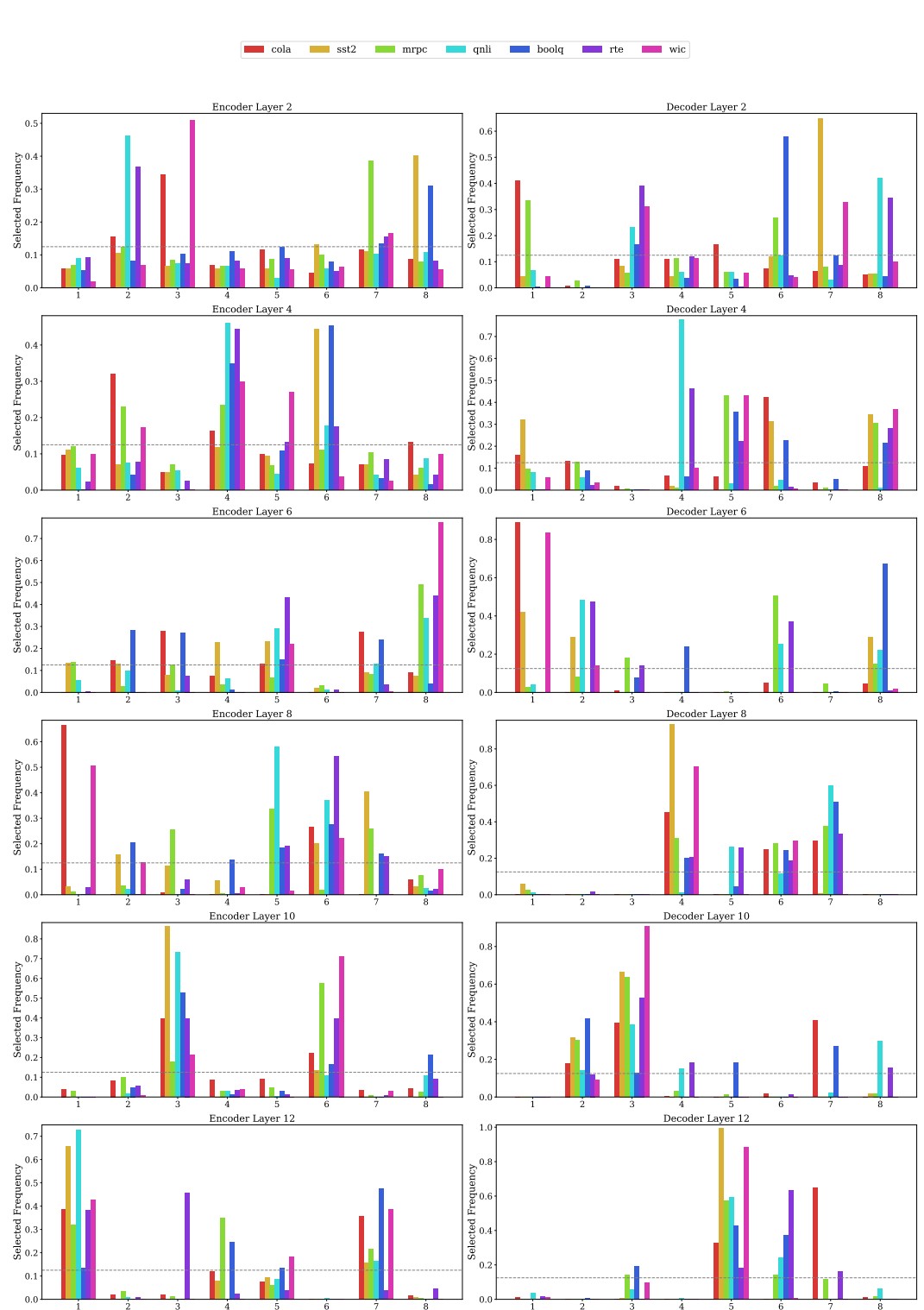

Figure 8: Selected frequency distribution of experts in each MoE layer for the model with **8** experts.

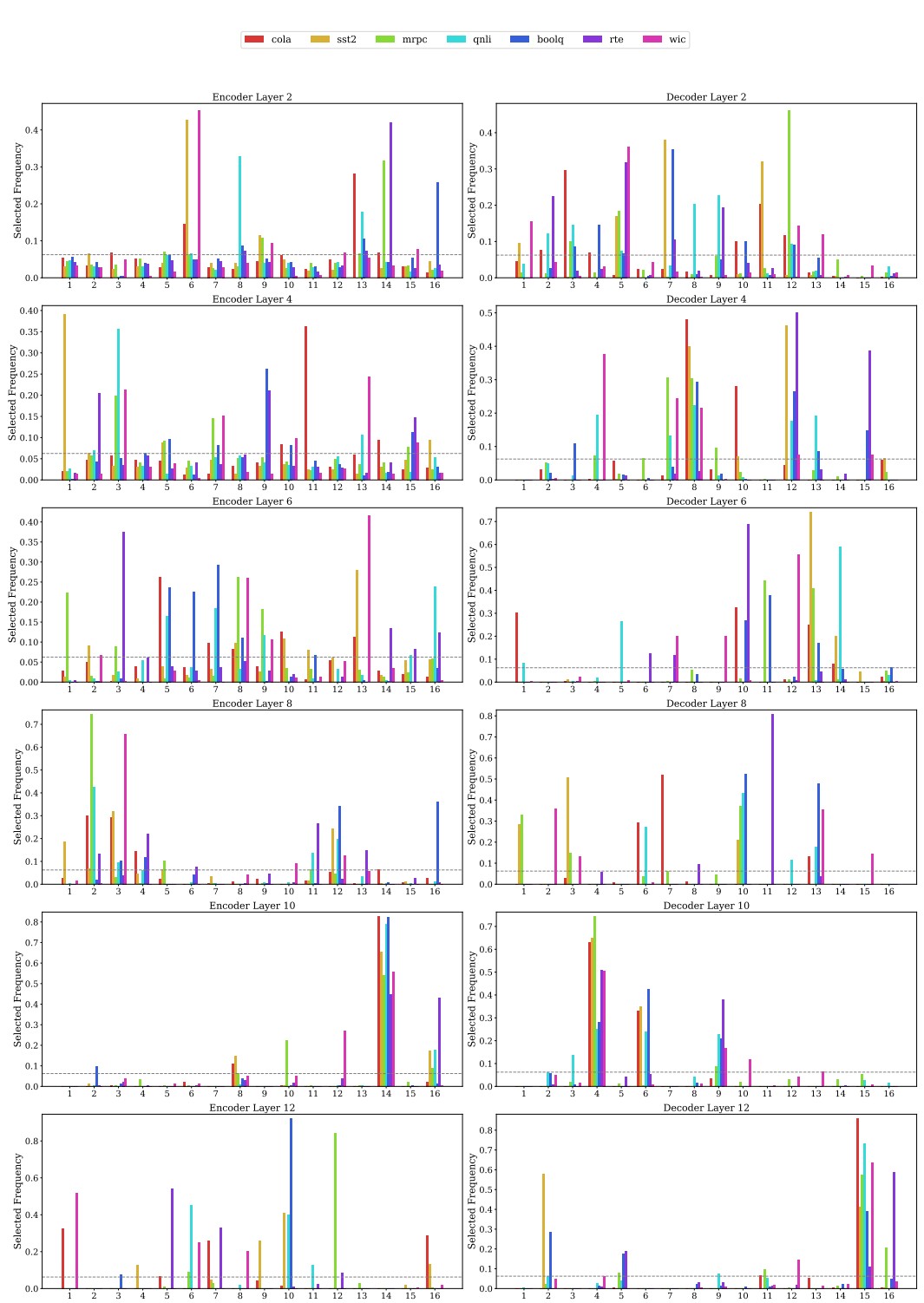

Figure 9: Selected frequency distribution of experts in each MoE layer for the model with **16** experts.

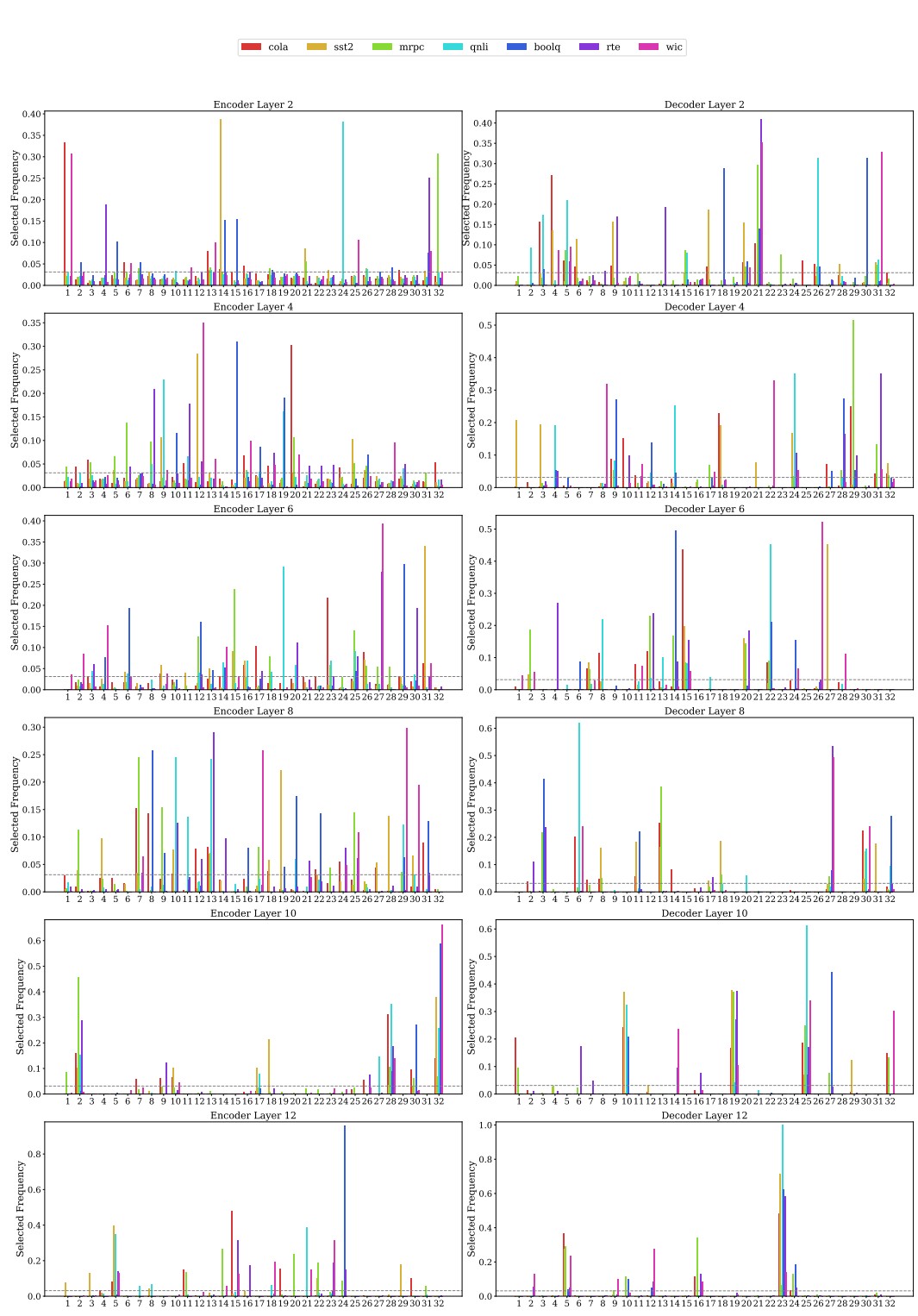

Figure 10: Selected frequency distribution of experts in each MoE layer for the model with **32** experts.

## C    DETAILS OF THE DATASET

Tab. 8 provides an overview of the datasets used in our experiments. We evaluate our approach on the GLUE and SuperGLUE benchmarks, which consist of various natural language understanding tasks, as well as the CNNDM dataset for summarization. For each dataset, we report the size of the training and test sets, the domain of the dataset, and the average length of the input sentences.

Table 8: Summary of the datasets used in our experiments. The table includes the corpus name, training and test set sizes, domain, and average sentence length (Avg. Len.). The GLUE and Super-GLUE benchmarks contain various natural language understanding tasks, while CNNDM is used for summarization.

| Benchmark | Dataset | Train | Test | Domain | Avg. Len. |
|-----------|---------|-------|------|--------|-----------|
| GLUE | CoLA | 8.5k | 1k | Misc. | 14.50 |
| | SST-2 | 67k | 1.8k | Movie reviews | 32.56 |
| | MRPC | 3.7k | 1.7k | News | 68.37 |
| | QNLI | 105k | 5.4k | Wikipedia | 60.25 |
| SuperGLUE | BoolQ | 9427 | 3245 | Google queries, Wikipedia | 157.40 |
| | RTE | 2490 | 3000 | News, Wikipedia | 79.08 |
| | WiC | 5428 | 1400 | WordNet, VerbNet, Wiktionary | 34.04 |
| - | CNNDM | 287.1k | 11.5k | News, Mail | 257.99 |

## D    HYPERPARAMETERS

Tab. 9 summarizes the hyperparameters used in our experiments for fine-tuning the models on the various datasets. The table includes the number of fine-tuning epochs, learning rate (lr.), whether BF16 precision is used, and the batch size for each dataset. The settings for GLUE and SuperGLUE benchmarks are tuned to ensure optimal performance across natural language understanding tasks. At the same time, the CNNDM dataset used for summarization requires a different configuration, particularly a larger learning rate and a smaller batch size due to the dataset's complexity.

Table 9: Summary of the hyperparameters used for fine-tuning on various datasets. The table lists the number of epochs, learning rate (lr.), BF16 precision usage, and batch size for each dataset. Different configurations are used for the GLUE, SuperGLUE benchmarks, and the CNNDM summarization dataset to ensure optimal performance across tasks.

| Benchmark | Dataset | # of Epochs | lr. | BF16 | Batch Size |
|-----------|---------|-------------|-----|------|-----------|
| GLUE | CoLA | 10 | 5e-5 | True | 32 |
| | SST-2 | 5 | 5e-5 | True | 32 |
| | MRPC | 10 | 5e-5 | True | 32 |
| | QNLI | 5 | 5e-5 | True | 32 |
| SuperGLUE | BoolQ | 10 | 5e-5 | True | 32 |
| | RTE | 15 | 5e-5 | True | 32 |
| | WiC | 15 | 5e-5 | True | 32 |
| - | CNNDM | 1 | 1e-4 | True | 16 |

## E    FINE-TUNING TIME COST

Tab. 10 presents the fine-tuning time required for the Switch Transformer across various downstream tasks, categorized by the number of experts utilized. The table includes results for the GLUE and SuperGLUE benchmarks, as well as the CNNDM dataset for summarization.

Table 10: Fine-tuning time (in hours) for the Switch Transformer across various downstream tasks, grouped by the number of experts.

| Benchmark | Dataset | 32 Experts | 16 Experts | 8 Experts | 4 Experts | 2 Experts | 1 Experts |
|---|---|---|---|---|---|---|---|
| GLUE | CoLA | 0.5 hr | 0.3 hr | 0.2 hr | 0.2 hr | 0.1 hr | 0.1 hr |
| | SST-2 | 2.0 hr | 1.3 hr | 0.9 hr | 0.7 hr | 0.6 hr | 0.5 hr |
| | MRPC | 0.3 hr | 0.2 hr | 0.1 hr | 0.1 hr | 0.1 hr | 0.1 hr |
| | QNLI | 3.2 hr | 2.4 hr | 1.8 hr | 1.4 hr | 1.2 hr | 1.1 hr |
| SuperGLUE | BoolQ | 1.0 hr | 0.7 hr | 0.5 hr | 0.4 hr | 0.3 hr | 0.3 hr |
| | RTE | 0.5 hr | 0.3 hr | 0.2 hr | 0.2 hr | 0.1 hr | 0.1 hr |
| | WiC | 0.8 hr | 0.6 hr | 0.5 hr | 0.4 hr | 0.3 hr | 0.3 hr |
| - | CNNDM | 4.2 hr | 3.3 hr | 2.6 hr | 2.2 hr | 1.9 hr | 1.8 hr |

## F  RESULT WITHOUT FINE-TUNING

Tab. 11 and 12 summarize the performance of the Switch Transformer with 64 experts and its reduced configurations using ExpertZIP without fine-tuning. Tab. 11 shows the results across various tasks in GLUE and SuperGLUE benchmarks, along with the average score degradation as the number of experts decreases. Tab. 12 highlights the ROUGE scores on the CNNDM summarization task under similar settings.

Table 11: Performance of the Switch Transformer (64 experts) and ExpertZIP configurations (with fewer experts) on GLUE and SuperGLUE benchmarks without fine-tuning.

| Method | # of Experts | CoLA | SST2 | MRPC | QNLI | BoolQ | RTE | WiC | Average ↑ |
|---|---|---|---|---|---|---|---|---|---|
| SwitchT. | 64 | 82.65 | 94.72 | 84.31 | 91.60 | 71.71 | 68.59 | 56.74 | 78.62 |
| ExpertZIP | 32 | 77.18 | 92.89 | 82.60 | 90.01 | 69.97 | 67.87 | 57.68 | 76.89 ( -2.25%) |
| | 16 | 58.87 | 85.67 | 72.79 | 86.75 | 64.43 | 64.98 | 56.43 | 69.99 (-12.33%) |
| | 8 | 57.14 | 78.10 | 72.55 | 85.65 | 63.12 | 64.26 | 55.17 | 68.00 (-15.62%) |
| | 4 | 55.61 | 75.80 | 71.57 | 85.12 | 63.91 | 63.18 | 55.17 | 67.19 (-17.01%) |
| | 2 | 54.83 | 74.31 | 70.59 | 85.15 | 63.09 | 62.82 | 55.49 | 66.61 (-18.03%) |
| | 1 | 53.02 | 70.87 | 70.59 | 84.94 | 63.33 | 61.73 | 54.70 | 65.45 (-20.12%) |

Table 12: Performance of the Switch Transformer (64 experts) and ExpertZIP configurations (with fewer experts) on CNNDM without fine-tuning.

| # of Experts | ROUGE-1 | ROUGE-2 | ROUGE-L |
|---|---|---|---|
| 64 | 36.19 | 16.46 | 27.44 |
| 32 | 32.42 | 13.09 | 24.24 |
| 16 | 31.55 | 12.59 | 23.62 |
| 8 | 31.40 | 12.18 | 22.73 |
| 4 | 25.36 | 8.69 | 19.14 |
| 2 | 25.32 | 8.85 | 19.04 |
| 1 | 24.70 | 7.68 | 19.04 |

## G  EXPERTZIP WITH OTHER MoE MODEL

Tab. 13 and 14 present additional experimental results to evaluate the robustness of ExpertZIP against the DeepSeek-MoE model Dai et al. (2024). The experiments span a variety of tasks from GLUE, SuperGLUE, and reasoning benchmarks.

These results highlight the ability of ExpertZIP to perform effectively with fewer experts and limited data usage, demonstrating its adaptability and reliability in diverse scenarios. ExpertZIP showcases its potential as a resource-efficient alternative to larger MoE configurations by reducing the number of experts while maintaining strong performance.

Table 13: Performance comparison between the original DeepSeekMoE model and the ExpertZIP model across various GLUE and SuperGLUE tasks on the RedPajama-1B dataset.

| Method | # of Experts | % of Dataset | CoLA | SST2 | MRPC | QNLI | BoolQ | RTE | WiC | Average ↑ |
|--------|-------------|-------------|------|------|------|------|-------|-----|-----|-----------|
| DeepSeekMoE | 64 | | 67.50 | 62.73 | 81.29 | 49.37 | 72.48 | 62.45 | 50.78 | 63.80 |
| ExpertZIP | 8 | 1% | 58.17 | 57.46 | 75.69 | 50.17 | 67.22 | 59.32 | 50.78 | 59.83 (-6.22%) |
| | | 5% | 60.29 | 59.85 | 77.34 | 50.23 | 68.59 | 60.15 | 51.23 | 61.10 (-4.23%) |
| | | 10% | 61.35 | 61.78 | 78.53 | 51.38 | 69.29 | 60.03 | 51.59 | 61.99 (-2.84%) |

Table 14: Performance comparison between the original DeepSeekMoE model and the ExpertZIP model across diverse reasoning and commonsense tasks on the RedPajama-1B dataset.

| Method | # of Experts | % of Dataset | MMLU | HellaS | PIQA | Arc-e | Arc-c | MathQA | WinoG | Average ↑ |
|--------|-------------|-------------|------|--------|------|-------|-------|--------|-------|-----------|
| DeepSeekMoE | 64 | | 37.81 | 77.38 | 80.03 | 76.05 | 48.12 | 31.76 | 70.48 | 60.23 |
| ExpertZIP | 8 | 1% | 30.18 | 70.76 | 75.03 | 74.19 | 45.72 | 27.47 | 63.04 | 55.20 (-8.35%) |
| | | 5% | 32.25 | 72.59 | 77.68 | 75.83 | 47.34 | 28.90 | 65.29 | 57.13 (-5.15%) |
| | | 10% | 34.91 | 74.36 | 78.52 | 75.35 | 47.26 | 29.18 | 68.23 | 58.26 (-3.27%) |

# H  EXPERTZIP WITH OTHER TASKS

To further validate the versatility of ExpertZIP, we evaluate its performance on additional tasks: WinoGrandeSakaguchi et al. (2021), WikiQAYang et al. (2015), and SQuAD Rajpurkar (2016). These tasks test the model's reasoning, question-answering, and comprehension capabilities. For evaluation, WinoGrande and WikiQA are measured in terms of accuracy, while SQuAD is evaluated using the *exact-match*. Tab. 15 compares the results of Switch Transformer (64 experts) with various configurations of ExpertZIP using fewer experts. The results demonstrate that ExpertZIP maintains competitive performance across these tasks, even with a significantly reduced number of experts, underscoring its robustness and efficiency.

Table 15: Performance of ExpertZIP across additional tasks: WinoGrande, WikiQA, and SQuAD, compared to the full Switch Transformer model with 64 experts.

| Method | # of Experts | WinoGrande | WikiQA | SQuAD |
|--------|-------------|-----------|--------|-------|
| SwitchT. | 64 | 62.98 | 95.61 | 65.81 |
| ExpertZIP | 32 | 62.18 | 95.32 | 65.39 |
| | 16 | 61.96 | 95.43 | 65.66 |
| | 8 | 61.78 | 95.10 | 65.41 |
| | 4 | 61.34 | 95.07 | 65.01 |
| | 2 | 61.67 | 94.95 | 64.97 |
| | 1 | 61.23 | 94.95 | 64.83 |

# I  DETAILS OF EXPERIMENT RESULTS

In this section, we provide a detailed overview of the experiment results. The number of experts is progressively reduced using various approaches, and performance metrics, including accuracy or ROUGE scores, are reported for each configuration.

Tab. 16 and 28 show the baseline results of fine-tuning the original Switch Transformers pre-trained weights without applying expert fusion techniques. This serves as a reference point to compare the impact of applying ExpertZIP.

The results of applying ExpertZIP with different fusion strategies are detailed in the subsequent tables. Tab. 29 shows the results of reducing the number of experts using a weighted-based approach, where the number of experts is progressively halved. Likewise, Tab. 17 and 30 display the results for an average-based fusion approach, also halving the number of experts at each step. Similarly, Tab. 18 and 31 present the results using a large-based fusion strategy.

The max-min and max-max approaches to determining how experts should be fused are shown in Tab. 19 and 21 for classification tasks, and Tab. 32 and 34 for summarization tasks. Additionally, Tab. 20 and 33 present the outcomes of using a random fusion approach.

For the experiments starting with fewer experts, the results are presented in Tab. 22, 23, 24, and 25. These experiments begin with 16 experts and gradually reduce the number of experts using different reduction strategies. Tab. 22 shows the results of gradually reducing one expert at a time (constant 1), while Tab. 23, 24, and 25 show the performance when reducing the number of experts by factors of 4, 8, and 16, respectively.

Lastly, for experiments starting with 32 and 16 experts, Tab. 26 and 27 show the results of halving the number of experts using the weight fusion approach on classification tasks. Similarly, the results for ExpertZIP applied to the summarization task, starting with 32 and 16 experts, are presented in Tab. 35 and 36, respectively.

Table 16: Results of fine-tuning the original Switch Transformers pre-trained weights on different classification tasks.

| # of Experts | CoLA | SST2 | MRPC | QNLI | BoolQ | RTE | WiC | Average ↑ |
|---|---|---|---|---|---|---|---|---|
| 64 | 82.65 | 94.72 | 84.31 | 91.60 | 71.71 | 68.59 | 56.74 | 78.62 |
| 32 | 80.92 | 94.04 | 85.78 | 89.77 | 69.30 | 63.54 | 57.84 | 77.31 |
| 16 | 82.07 | 91.06 | 83.09 | 88.30 | 66.91 | 63.54 | 54.23 | 75.60 |
| 8 | 74.21 | 92.09 | 79.66 | 85.94 | 65.99 | 59.93 | 52.66 | 72.93 |

Table 17: Results of fine-tuning after applying ExpertZIP, starting with **64** experts and progressively halving the number of experts through weight fusion using an **average-based** approach on different classification tasks.

| # of Experts | CoLA | SST2 | MRPC | QNLI | BoolQ | RTE | WiC | Average ↑ |
|---|---|---|---|---|---|---|---|---|
| 64 | 82.65 | 94.72 | 84.31 | 91.60 | 71.71 | 68.59 | 56.74 | 78.62 |
| 32 | 80.92 | 93.81 | 84.80 | 90.48 | 71.71 | 68.23 | 56.90 | 78.12 |
| 16 | 79.19 | 92.09 | 84.31 | 89.88 | 72.81 | 69.31 | 56.11 | 77.67 |
| 8 | 76.70 | 91.63 | 83.82 | 89.64 | 72.57 | 68.23 | 55.96 | 76.94 |
| 4 | 75.26 | 91.86 | 83.82 | 89.44 | 72.91 | 68.23 | 56.30 | 76.83 |
| 2 | 75.07 | 91.63 | 83.82 | 89.88 | 72.57 | 67.51 | 56.27 | 76.68 |
| 1 | 74.78 | 90.94 | 84.07 | 89.58 | 72.84 | 67.51 | 56.58 | 76.61 |

Table 18: Results of fine-tuning after applying ExpertZIP, starting with **64** experts and progressively halving the number of experts through weight fusion using a **large-based** approach on different classification tasks.

| # of Experts | CoLA | SST2 | MRPC | QNLI | BoolQ | RTE | WiC | Average ↑ |
|---|---|---|---|---|---|---|---|---|
| 64 | 82.65 | 94.72 | 84.31 | 91.60 | 71.71 | 68.59 | 56.74 | 78.62 |
| 32 | 81.21 | 93.69 | 83.82 | 90.61 | 71.59 | 68.59 | 55.33 | 77.83 |
| 16 | 78.81 | 91.86 | 84.80 | 89.84 | 70.95 | 66.43 | 55.49 | 76.88 |
| 8 | 76.89 | 91.28 | 83.82 | 89.95 | 71.01 | 64.26 | 55.33 | 76.08 |
| 4 | 75.74 | 91.51 | 82.35 | 89.18 | 71.25 | 63.18 | 55.17 | 75.48 |
| 2 | 75.17 | 90.60 | 82.35 | 89.33 | 71.65 | 63.90 | 54.23 | 75.32 |
| 1 | 74.50 | 90.25 | 81.86 | 89.46 | 71.44 | 63.90 | 55.33 | 75.25 |

Table 19: Results of fine-tuning. Starting with **64** experts and progressively halving the number of experts using the **max-min** approach to determine how experts should be fused on different classification tasks.

| # of Experts | CoLA | SST2 | MRPC | QNLI | BoolQ | RTE | WiC | Average ↑ |
|---|---|---|---|---|---|---|---|---|
| 64 | 82.65 | 94.72 | 84.31 | 91.60 | 71.71 | 68.59 | 56.74 | 78.62 |
| 32 | 81.02 | 94.04 | 85.54 | 90.32 | 71.80 | 67.51 | 56.74 | 78.14 |
| 16 | 79.58 | 92.32 | 85.05 | 90.01 | 72.29 | 67.51 | 55.80 | 77.51 |
| 8 | 76.61 | 91.06 | 82.84 | 89.88 | 72.72 | 67.87 | 55.64 | 76.66 |
| 4 | 76.13 | 90.83 | 83.09 | 89.60 | 73.00 | 67.51 | 55.96 | 76.59 |
| 2 | 75.74 | 90.48 | 83.33 | 89.05 | 73.27 | 66.79 | 56.27 | 76.42 |
| 1 | 75.17 | 90.71 | 83.09 | 89.47 | 73.27 | 67.15 | 55.80 | 76.38 |

Table 20: Results of fine-tuning. Starting with **64** experts and progressively halving the number of experts using the **random** approach to determine how experts should be fused on different classification tasks.

| # of Experts | CoLA | SST2 | MRPC | QNLI | BoolQ | RTE | WiC | Average ↑ |
|---|---|---|---|---|---|---|---|---|
| 64 | 82.65 | 94.72 | 84.31 | 91.60 | 71.71 | 68.59 | 56.74 | 78.62 |
| 32 | 80.15 | 93.12 | 84.80 | 90.76 | 72.23 | 68.59 | 55.33 | 77.85 |
| 16 | 78.72 | 91.74 | 85.54 | 90.43 | 72.63 | 67.51 | 55.33 | 77.41 |
| 8 | 76.51 | 91.06 | 82.84 | 89.42 | 72.63 | 66.79 | 55.64 | 76.41 |
| 4 | 74.59 | 90.25 | 82.84 | 89.29 | 72.97 | 67.51 | 56.27 | 76.25 |
| 2 | 74.69 | 90.48 | 82.11 | 89.24 | 72.29 | 68.59 | 56.27 | 76.24 |
| 1 | 74.98 | 90.83 | 82.35 | 89.88 | 73.03 | 66.06 | 56.11 | 76.18 |

Table 21: Results of fine-tuning. Starting with **64** experts and progressively halving the number of experts using the **max-max** approach to determine how experts should be fused on different classification tasks.

| # of Experts | CoLA | SST2 | MRPC | QNLI | BoolQ | RTE | WiC | Average ↑ |
|---|---|---|---|---|---|---|---|---|
| 64 | 82.65 | 94.72 | 84.31 | 91.60 | 71.71 | 68.59 | 56.74 | 78.62 |
| 32 | 78.62 | 92.20 | 84.56 | 91.09 | 73.46 | 67.15 | 56.43 | 77.64 |
| 16 | 75.26 | 90.94 | 83.33 | 90.12 | 73.06 | 66.06 | 56.27 | 76.43 |
| 8 | 74.40 | 90.48 | 82.35 | 89.82 | 72.75 | 66.43 | 55.49 | 75.96 |
| 4 | 74.21 | 90.48 | 82.60 | 89.11 | 72.14 | 66.43 | 55.33 | 75.76 |
| 2 | 73.73 | 90.25 | 82.60 | 89.11 | 72.20 | 66.06 | 55.33 | 75.61 |
| 1 | 73.63 | 90.71 | 82.11 | 89.20 | 72.02 | 66.06 | 55.02 | 75.54 |

Table 22: Results of fine-tuning after applying ExpertZIP, starting with **16** experts and gradually reducing the number of experts one at a time (**constant 1**) using a **weighted-based** approach to fuse the weights on different classification tasks.

| # of Experts | CoLA | SST2 | MRPC | QNLI | BoolQ | RTE | WiC | Average ↑ |
|---|---|---|---|---|---|---|---|---|
| 16 | 78.24 | 92.32 | 84.56 | 90.08 | 72.72 | 69.68 | 56.43 | 77.72 |
| 15 | 78.43 | 93.00 | 85.29 | 90.23 | 73.24 | 69.31 | 55.33 | 77.83 |
| 14 | 77.85 | 92.66 | 86.03 | 90.54 | 73.67 | 68.59 | 55.80 | 77.88 |
| 13 | 77.66 | 92.32 | 85.54 | 90.37 | 73.64 | 68.59 | 55.49 | 77.66 |
| 12 | 77.85 | 92.32 | 84.80 | 90.24 | 73.82 | 68.59 | 55.80 | 77.63 |
| 11 | 77.66 | 92.43 | 85.05 | 90.43 | 73.94 | 69.31 | 55.64 | 77.78 |
| 10 | 78.43 | 92.78 | 84.31 | 90.35 | 73.94 | 69.31 | 54.55 | 77.67 |
| 9 | 77.37 | 92.43 | 84.56 | 90.35 | 74.19 | 69.31 | 55.80 | 77.72 |
| 8 | 77.85 | 92.55 | 84.31 | 90.10 | 73.61 | 69.31 | 55.80 | 77.65 |
| 7 | 78.04 | 92.66 | 84.31 | 90.37 | 73.67 | 69.68 | 55.64 | 77.77 |
| 6 | 77.56 | 92.09 | 84.56 | 90.15 | 73.21 | 69.68 | 56.74 | 77.71 |
| 5 | 77.28 | 92.20 | 84.07 | 89.91 | 74.43 | 69.31 | 56.27 | 77.64 |
| 4 | 77.28 | 92.20 | 84.07 | 90.06 | 73.70 | 69.68 | 57.05 | 77.72 |
| 3 | 77.66 | 92.55 | 83.82 | 90.06 | 73.70 | 68.23 | 57.99 | 77.72 |
| 2 | 76.80 | 92.09 | 85.05 | 89.93 | 73.91 | 69.31 | 57.21 | 77.76 |
| 1 | 77.09 | 92.09 | 84.80 | 90.30 | 73.52 | 68.95 | 57.05 | 77.69 |

Table 23: Results of fine-tuning after applying ExpertZIP, starting with **16** experts and reducing the number of experts by a factor of 4 (**speed 4**) using a **weighted-based** approach to fuse the weights on different classification tasks.

| # of Experts | CoLA | SST2 | MRPC | QNLI | BoolQ | RTE | WiC | Average ↑ |
|---|---|---|---|---|---|---|---|---|
| 16 | 78.24 | 92.32 | 84.56 | 90.08 | 72.72 | 69.68 | 56.43 | 77.72 |
| 4 | 75.46 | 91.06 | 83.82 | 89.38 | 72.57 | 68.95 | 56.90 | 76.88 |
| 1 | 74.40 | 91.28 | 83.33 | 89.24 | 73.30 | 68.23 | 56.90 | 76.67 |

Table 24: Results of fine-tuning after applying ExpertZIP, starting with **16** experts and reducing the number of experts by a factor of 8 (**speed 8**) using a **weighted-based** approach to fuse the weights on different classification tasks.

| # of Experts | CoLA | SST2 | MRPC | QNLI | BoolQ | RTE | WiC | Average ↑ |
|---|---|---|---|---|---|---|---|---|
| 16 | 78.24 | 92.32 | 84.56 | 90.08 | 72.72 | 69.68 | 56.43 | 77.72 |
| 2 | 74.21 | 91.40 | 84.31 | 89.09 | 72.39 | 69.31 | 55.02 | 76.53 |
| 1 | 74.68 | 91.51 | 83.58 | 89.11 | 72.23 | 68.95 | 55.49 | 76.51 |

Table 25: Results of fine-tuning after applying ExpertZIP, starting with **16** experts and reducing the number of experts by a factor of 16 (**speed 16**) using a **weighted-based** approach to fuse the weights on different classification tasks.

| # of Experts | CoLA | SST2 | MRPC | QNLI | BoolQ | RTE | WiC | Average ↑ |
|---|---|---|---|---|---|---|---|---|
| 16 | 78.24 | 92.32 | 84.56 | 90.08 | 72.72 | 69.68 | 56.43 | 77.72 |
| 1 | 74.21 | 90.37 | 83.33 | 89.55 | 72.29 | 68.23 | 56.90 | 76.41 |

Table 26: Results of fine-tuning after applying ExpertZIP, starting with **32** experts and progressively halving the number of experts through weight fusion using an **weighted-based** approach on different classification tasks.

| # of Experts | CoLA | SST2 | MRPC | QNLI | BoolQ | RTE | WiC | Average ↑ |
|---|---|---|---|---|---|---|---|---|
| 32 | 80.92 | 94.04 | 85.78 | 89.77 | 69.30 | 63.54 | 57.84 | 77.31 |
| 16 | 75.26 | 91.28 | 84.31 | 89.97 | 72.97 | 68.95 | 55.49 | 76.89 |
| 8 | 74.40 | 90.37 | 83.82 | 89.24 | 73.30 | 68.23 | 56.90 | 76.61 |

Table 27: Results of fine-tuning after applying ExpertZIP, starting with **16** experts and progressively halving the number of experts through weight fusion using an **weighted-based** approach on different classification tasks.

| # of Experts | CoLA | SST2 | MRPC | QNLI | BoolQ | RTE | WiC | Average ↑ |
|---|---|---|---|---|---|---|---|---|
| 16 | 82.07 | 91.06 | 83.09 | 88.30 | 66.91 | 63.54 | 54.23 | 75.60 |
| 8 | 78.14 | 90.25 | 82.60 | 90.23 | 68.04 | 60.65 | 55.64 | 75.08 |

Table 28: Results of fine-tuning the original Switch Transformers pre-trained weights on the CN-NDM summarization task.

| # of Experts | ROUGE-1 | ROUGE-2 | ROUGE-L |
|---|---|---|---|
| 64 | 36.19 | 16.46 | 27.44 |
| 32 | 35.86 | 16.58 | 28.07 |
| 16 | 34.34 | 14.40 | 26.11 |
| 8 | 32.81 | 13.36 | 24.25 |

Table 29: Results of fine-tuning after applying ExpertZIP, starting with **64** experts and progressively halving the number of experts through weight fusion using a **weighted-based** approach on the CN-NDM summarization task.

| # of Experts | ROUGE-1 | ROUGE-2 | ROUGE-L |
|---|---|---|---|
| 64 | 36.19 | 16.46 | 27.44 |
| 32 | 35.88 | 15.90 | 27.61 |
| 16 | 36.05 | 16.56 | 27.32 |
| 8 | 35.35 | 15.43 | 27.28 |
| 4 | 35.28 | 15.92 | 27.37 |
| 2 | 35.56 | 16.34 | 26.66 |
| 1 | 35.07 | 15.27 | 26.18 |

Table 30: Results of fine-tuning after applying ExpertZIP, starting with **64** experts and progressively halving the number of experts through weight fusion using a **average-based** approach on the CN-NDM summarization task.

| # of Experts | ROUGE-1 | ROUGE-2 | ROUGE-L |
|---|---|---|---|
| 64 | 36.19 | 16.46 | 27.44 |
| 32 | 35.73 | 16.04 | 27.02 |
| 16 | 35.29 | 15.71 | 26.92 |
| 8 | 34.40 | 14.96 | 26.67 |
| 4 | 34.30 | 15.18 | 26.19 |
| 2 | 34.59 | 14.86 | 26.21 |
| 1 | 34.69 | 14.97 | 26.49 |

Table 31: Results of fine-tuning after applying ExpertZIP, starting with **64** experts and progressively halving the number of experts through weight fusion using a **large-based** approach on the CNNDM summarization task.

| # of Experts | ROUGE-1 | ROUGE-2 | ROUGE-L |
|---|---|---|---|
| 64 | 36.19 | 16.46 | 27.44 |
| 32 | 35.18 | 15.66 | 26.56 |
| 16 | 35.00 | 14.77 | 26.18 |
| 8 | 34.41 | 15.20 | 27.12 |
| 4 | 34.13 | 15.15 | 26.37 |
| 2 | 33.17 | 13.87 | 24.62 |
| 1 | 32.89 | 13.62 | 24.33 |

Table 32: Results of fine-tuning. Starting with **64** experts and progressively halving the number of experts using the **max-min** approach to determine how experts should be fused on the CNNDM summarization task.

| # of Experts | ROUGE-1 | ROUGE-2 | ROUGE-L |
|---|---|---|---|
| 64 | 36.19 | 16.46 | 27.44 |
| 32 | 35.61 | 15.35 | 27.30 |
| 16 | 34.97 | 14.91 | 26.06 |
| 8 | 34.82 | 15.12 | 26.60 |
| 4 | 35.07 | 15.13 | 26.40 |
| 2 | 34.85 | 15.61 | 26.73 |
| 1 | 34.75 | 15.01 | 26.29 |

Table 33: Results of fine-tuning. Starting with **64** experts and progressively halving the number of experts using the **random** approach to determine how experts should be fused on the CNNDM summarization task.

| # of Experts | ROUGE-1 | ROUGE-2 | ROUGE-L |
|---|---|---|---|
| 64 | 36.19 | 16.46 | 27.44 |
| 32 | 35.12 | 15.17 | 26.20 |
| 16 | 35.02 | 15.40 | 26.48 |
| 8 | 34.72 | 15.49 | 26.30 |
| 4 | 34.38 | 15.16 | 26.41 |
| 2 | 34.02 | 14.91 | 25.82 |
| 1 | 33.99 | 14.56 | 25.17 |

Table 34: Results of fine-tuning. Starting with **64** experts and progressively halving the number of experts using the **max-max** approach to determine how experts should be fused on the CNNDM summarization task.

| # of Experts | ROUGE-1 | ROUGE-2 | ROUGE-L |
|---|---|---|---|
| 64 | 36.19 | 16.46 | 27.44 |
| 32 | 35.31 | 15.80 | 26.85 |
| 16 | 34.67 | 15.40 | 26.41 |
| 8 | 34.43 | 15.21 | 26.56 |
| 4 | 34.53 | 15.26 | 26.47 |
| 2 | 34.09 | 14.81 | 26.35 |
| 1 | 33.73 | 14.02 | 25.21 |

Table 35: Results of fine-tuning after applying ExpertZIP, starting with **32** experts and progressively halving the number of experts through weight fusion using a **weighted-based** approach on the CNNDM summarization task.

| # of Experts | ROUGE-1 | ROUGE-2 | ROUGE-L |
|---|---|---|---|
| 32 | 35.86 | 16.58 | 28.07 |
| 16 | 34.70 | 15.81 | 26.95 |
| 8 | 34.26 | 13.89 | 25.20 |

Table 36: Results of fine-tuning after applying ExpertZIP, starting with **16** experts and progressively halving the number of experts through weight fusion using a **weighted-based** approach on the CNNDM summarization task.

| # of Experts | ROUGE-1 | ROUGE-2 | ROUGE-L |
|---|---|---|---|
| 16 | 34.34 | 14.40 | 26.11 |
| 8 | 33.40 | 14.65 | 25.29 |

