# OpenReview forum: "ExpertZIP: A Progressive Fusion Framework for Mixture-of-Experts Model Optimization through Huffman Tree Structures"
_ICLR.cc/2025/Conference — Submitted to ICLR 2025_

### Official Review · Reviewer_UUqs · 2024-10-26

**Soundness:** 2
**Presentation:** 3
**Contribution:** 2
**Rating:** 6
**Confidence:** 4

**Summary:**

This paper proposes a three-step pipeline to reduce the number of parameters and improve the inference time of popular MoE models, such as Switch Transformer and Mixtral 8x7B. First, it groups underutilized experts, then merges them using frequency-weighted averaging, and finally fine-tunes the model to recover accuracy.

**Strengths:**

- The analysis of expert utilization across different benchmarks strongly motivates this work.
- The figures clearly demonstrate the logical flow and effectively illustrate the main idea.

**Weaknesses:**

- The authors did not compare their method to existing baselines. Could the authors consider referencing [1] and [2]? The former performs expert merging on the Switch Transformer, and the latter prunes experts based on activation frequency.
- The authors did not clearly specify the model in the abstract and introduction, which may lead to confusion about the reported improvements, as different models may yield varying results.

[1] Li et al. Merge, Then Compress: Demystify Efficient SMoE with Hints from Its Routing Policy, ICLR 2024.\
[2] He et al. Demystifying the Compression of Mixture-of-Experts Through a Unified Framework, 2024.

**Questions:**

- For each task benchmark, is the fine-tuning process independent? More specifically, is the model evaluated on CoLA and SST-2 have exactly the same parameters? It seems that there are two models, one is fine-tuned on CoLA, and the other is fine-tuned on SST-2.
- Could the authors provide the time cost for fine-tuning and the accuracy after fusing without fine-tuning?
- Could the authors provide more details on the reasons for the decreased inference cost? Is it due to modifying the original router strategy from top-k to top-1 and then fine-tuning the model accordingly? Or is the cost reduction primarily due to time saved by loading experts into GPU memory?
- Could the authors specify which MoE model you use to analyze in Fig. 3?
- Could the authors explicitly compare the proposed method's performance and efficiency gains against the results reported in [1] and [2], highlighting any key differences in approach or outcomes?
- Could the authors add a description regarding the adopted MoE model architecture in the introduction and abstract sections?

---

> ### Author Response · Authors · 2024-11-16
> **Response to Reviewer UUqs's Comments**
>
> Thank you for your detailed review and constructive suggestions. We are grateful for your positive feedback on the logical motivation and visual clarity of our work. Below, we provide detailed responses to each of your comments and questions.
>
> > Comparison with Existing Baselines
>
> As shown in the table below, ExpertZIP demonstrates competitive performance while significantly reducing model size compared to pruning, quantization, and merging methods, including those proposed in [6] and [3]. Notably, the enhanced variant, ExpertZIP*, achieves the highest average score (77.69) with the lowest performance degradation (-1.18%), outperforming M-SMoE from [6] and showing greater robustness than pruning methods in [3]. These results highlight the effectiveness of ExpertZIP in maintaining model robustness while optimizing efficiency.
>
> | Category               | Method                  | Model Size | CoLA   | SST2   | MRPC   | QNLI   | BoolQ  | RTE    | WiC    | Average ↑            |
> |------------------------|-------------------------|------------|--------|--------|--------|--------|--------|--------|--------|----------------------|
> |                        | SwitchT. (64 expert)   | 3.79B      | 82.65  | 94.72  | 84.31  | 91.60  | 71.71  | 68.59  | 56.74  | 78.62               |
> | Pruning / Quantization | Task-Specific [1]      | 0.22B      | 70.25  | 86.43  | 75.76  | 80.24  | 67.24  | 65.43  | **57.21** | 71.79 (-8.69%)      |
> |                        | PS-MoE [2]             | 0.22B      | 72.14  | 87.18  | 74.38  | 83.25  | 70.75  | 64.97  | 56.43  | 72.73 (-7.49%)      |
> |                        | UV-MoE [3]             | 0.22B      | 71.25  | 88.47  | 80.23  | 79.67  | 71.71  | 65.43  | 56.89  | 73.38 (-6.67%)      |
> | Merging                | REPAIR [4]             | 0.52B      | 74.02  | 90.21  | 83.27  | 86.75  | 73.39  | 67.23  | 58.42  | 76.18 (-3.10%)      |
> |                        | ZipIt [5]              | 0.52B      | 73.98  | 91.78  | 82.58  | 87.90  | 72.14  | 66.18  | 57.05  | 75.94 (-3.41%)      |
> |                        | M-SMoE (1 expert) [6]  | 0.52B      | 75.18  | **92.43** | 83.53  | 88.94  | **74.29** | 68.59  | 56.43  | 77.06 (-1.98%)      |
> | Ours                   | ExpertZIP (1 expert)   | 0.22B      | 75.26  | 91.28  | 84.07  | 90.01  | 73.39  | 68.23  | 56.89  | 77.02 (-2.04%)      |
> |                        | ExpertZIP* (1 expert)  | 0.22B      | **77.09** | 92.09  | **84.80** | **90.30** | 73.52  | **68.95** | 57.05  | **77.69 (-1.18%)** |
>
>
>
> > Model Specification in Abstract and Introduction
>
> We have updated the manuscript to explicitly state that the Switch Transformer model was utilized in our experiments. This clarification ensures transparency and consistency across the paper.
>
> > Independence of Fine-Tuning Across Benchmarks
>
> When using the Switch Transformer, the fine-tuning process for each downstream task is performed independently due to its architecture, which is not inherently generalized. However, for the Mixtral 8x7B model, the same model can be evaluated across multiple benchmarks without independent fine-tuning. Both cases are discussed in the revised Discussion section, which includes results for both models.

---

> > ### Author Response · Authors · 2024-11-16
> > **Response to Reviewer UUqs's Comments**
> >
> > > Time Cost for Fine-Tuning and Accuracy After Fusion Without Fine-Tuning
> >
> > The fine-tuning time costs for various benchmarks have been added to Appendix E, as shown in the table below. Regarding accuracy after fusion without fine-tuning, our experiments suggest that merging experts without subsequent fine-tuning can degrade performance due to differences in the output distributions of individual experts. We strongly recommend fine-tuning the model post-merging to ensure the merged expert can learn accurate representations.
> >
> > | Benchmark     | Dataset  | 32 Experts | 16 Experts | 8 Experts | 4 Experts | 2 Experts | 1 Expert |
> > |---------------|----------|------------|------------|-----------|-----------|-----------|----------|
> > | GLUE          | CoLA     | 0.5 hr     | 0.3 hr     | 0.2 hr    | 0.2 hr    | 0.1 hr    | 0.1 hr   |
> > |               | SST-2    | 2.0 hr     | 1.3 hr     | 0.9 hr    | 0.7 hr    | 0.6 hr    | 0.5 hr   |
> > |               | MRPC     | 0.3 hr     | 0.2 hr     | 0.1 hr    | 0.1 hr    | 0.1 hr    | 0.1 hr   |
> > |               | QNLI     | 3.2 hr     | 2.4 hr     | 1.8 hr    | 1.4 hr    | 1.2 hr    | 1.1 hr   |
> > | SuperGLUE     | BoolQ    | 1.0 hr     | 0.7 hr     | 0.5 hr    | 0.4 hr    | 0.3 hr    | 0.3 hr   |
> > |               | RTE      | 0.5 hr     | 0.3 hr     | 0.2 hr    | 0.2 hr    | 0.1 hr    | 0.1 hr   |
> > |               | WiC      | 0.8 hr     | 0.6 hr     | 0.5 hr    | 0.4 hr    | 0.3 hr    | 0.3 hr   |
> > | -             | CNNDM    | 4.2 hr     | 3.3 hr     | 2.6 hr    | 2.2 hr    | 1.9 hr    | 1.8 hr   |
> >
> >
> > | Method     | # of Experts | CoLA   | SST2   | MRPC   | QNLI   | BoolQ  | RTE    | WiC    | Average ↑            |
> > |------------|--------------|--------|--------|--------|--------|--------|--------|--------|----------------------|
> > | SwitchT.   | 64           | 82.65  | 94.72  | 84.31  | 91.60  | 71.71  | 68.59  | 56.74  | 78.62               |
> > | ExpertZIP  | 32           | 77.18  | 92.89  | 82.60  | 90.01  | 69.97  | 67.87  | 57.68  | 76.89 (-2.25%)      |
> > |            | 16           | 58.87  | 85.67  | 72.79  | 86.75  | 64.43  | 64.98  | 56.43  | 69.99 (-12.33%)     |
> > |            | 8            | 57.14  | 78.10  | 72.55  | 85.65  | 63.12  | 64.26  | 55.17  | 68.00 (-15.62%)     |
> > |            | 4            | 55.61  | 75.80  | 71.57  | 85.12  | 63.91  | 63.18  | 55.17  | 67.19 (-17.01%)     |
> > |            | 2            | 54.83  | 74.31  | 70.59  | 85.15  | 63.09  | 62.82  | 55.49  | 66.61 (-18.03%)     |
> > |            | 1            | 53.02  | 70.87  | 70.59  | 84.94  | 63.33  | 61.73  | 54.70  | 65.45 (-20.12%)     |
> >
> >
> >
> > | # of Experts | ROUGE-1 | ROUGE-2 | ROUGE-L |
> > |--------------|---------|---------|---------|
> > | 64           | 36.19   | 16.46   | 27.44   |
> > | 32           | 32.42   | 13.09   | 24.24   |
> > | 16           | 31.55   | 12.59   | 23.62   |
> > | 8            | 31.40   | 12.18   | 22.73   |
> > | 4            | 25.36   | 8.69    | 19.14   |
> > | 2            | 25.32   | 8.85    | 19.04   |
> > | 1            | 24.70   | 7.68    | 19.04   |
> >
> >
> > > Reason for Decreased Inference Cost
> >
> > The reduction in inference cost is primarily due to the reduced number of experts, which accelerates token processing. In sequential processing, fewer experts reduce traversal time, as discussed in [7]. Similarly, for parallel processing, the fixed GPU memory necessitates longer processing times for more experts due to frequent device movement or removal.
> >
> > > MoE Model Used in Figure 3
> >
> > In Figure 3, we utilized the Switch Transformer model with 8 experts. This information has been added to the figure caption for clarity.
> >
> > The above results are included in the revised version of the manuscript.
> >
> > Thank you once again for your valuable feedback. Your suggestions have significantly improved the clarity and comprehensiveness of our manuscript. We welcome any further questions or comments.
> >
> > [1] Chen, Tianyu, et al. "Task-specific expert pruning for sparse mixture-of-experts." arXiv preprint arXiv:2206.00277 (2022).
> >
> > [2] Lu, Xudong, et al. "Not All Experts are Equal: Efficient Expert Pruning and Skipping for Mixture-of-Experts Large Language Models." arXiv preprint arXiv:2402.14800 (2024).
> >
> > [3] He, Shwai, et al. "Demystifying the Compression of Mixture-of-Experts Through a Unified Framework." arXiv preprint arXiv:2406.02500 (2024).
> >
> > [4] Jordan, Keller, et al. "Repair: Renormalizing permuted activations for interpolation repair." arXiv preprint arXiv:2211.08403 (2022).
> >
> > [5] Stoica, George, et al. "Zipit! merging models from different tasks without training." arXiv preprint arXiv:2305.03053 (2023).
> >
> > [6] Li, Pingzhi, et al. "Merge, then compress: Demystify efficient SMoe with hints from its routing policy." arXiv preprint arXiv:2310.01334 (2023).
> >
> > [7] Du, Zhixu, et al. "SiDA: Sparsity-Inspired Data-Aware Serving for Efficient and Scalable Large Mixture-of-Experts Models." Proceedings of Machine Learning and Systems 6 (2024): 224-238.

---

> > > ### Author Response · Authors · 2024-11-22
> > > **Follow Up Reminder**
> > >
> > > Dear Reviewer UUqs,
> > >
> > > We greatly appreciate your valuable feedback and have carefully considered your comments in our responses. We kindly wish to confirm if our responses have sufficiently addressed the concerns you raised.
> > >
> > > If you find our revisions satisfactory and think we have adequately addressed your concerns, we would really appreciate it if you could consider revisiting your initial evaluation. If any additional points require clarification or further adjustments, please do not hesitate to let us know.
> > >
> > > Thank you once again for your time and dedication in reviewing our work.
> > >
> > > Many thanks, Authors

---

> > > > ### Comment · Reviewer_UUqs · 2024-12-02
> > > >
> > > > I appreciate the authors' efforts in the rebuttal. Regarding the performance comparison, could the authors align the number of parameters and provide a comparison accordingly (e.g., ExpertZIP with 0.52B parameters or merging methods with 0.22B)?

---

> ### Author Response · Authors · 2024-12-02
> **Response to Reviewer UUqs's Comments**
>
> Thank you for your valuable feedback. The decision to use a 0.52B parameter configuration was based on the methodology outlined in M-SMoE [1], specifically referring to Table 2 in their paper. As stated in the caption, M-SMoE's experiments followed the practice in LLM-Pruner[2], which does not merge the first layer of the encoder. To ensure a fair comparison, we adopted the same experimental setup, allowing for direct and consistent benchmarking against their results. This alignment ensures that our adjustments and merging methodology are fairly evaluated under comparable conditions.
>
> Additionally, to address your concern about parameter alignment, we have provided results for M-SMoE when reduced to 0.22B parameters. These results (shown below) demonstrate that ExpertZIP outperforms M-SMoE under the 0.22B parameter setting, further highlighting the effectiveness of our approach:
>
> | Method                     | Model Size | CoLA   | SST2   | MRPC   | QNLI   | BoolQ  | RTE    | WiC    | Average ↑          |
> |----------------------------|------------|--------|--------|--------|--------|--------|--------|--------|--------------------|
> | SwitchT. (64 expert)       | 3.79B      | 82.65  | 94.72  | 84.31  | 91.60  | 71.71  | 68.59  | 56.74  | 78.62              |
> | M-SMoE (w/ 1st layer merge) | 0.22B | 74.11  | 91.28 | 84.31  | 89.75  | 73.64 | 67.51  | **57.21**  | 76.83 (-2.28%)     |
> | M-SMoE (wo/ 1st layer merge)  | 0.52B      | 75.18  | **92.43** | 83.53  | 88.94  | **74.29** | 68.59  | 56.43  | 77.06 (-1.98%)      |
> | ExpertZIP (1 expert)       | 0.22B      | 75.26  | 91.28  | 84.07  | 90.01  | 73.39  | 68.23  | 56.89  | 77.02 (-2.04%)     |
> | ExpertZIP* (1 expert)      | 0.22B      | **77.09** | 92.09  | **84.80** | **90.30** | 73.52  | **68.95** | 57.05  | **77.69 (-1.18%)** |
>
> We acknowledge the importance of testing under varying conditions and appreciate your suggestion. In future work, we plan to extend our experiments to include settings where the first layer of the encoder remains unmerged for a more comprehensive evaluation.
>
> Thank you for bringing this to our attention.
>
> [1] Li, Pingzhi, et al. "Merge, then compress: Demystify efficient SMoe with hints from its routing policy." arXiv preprint arXiv:2310.01334 (2023).
>
> [2] Ma, Xinyin, Gongfan Fang, and Xinchao Wang. "Llm-pruner: On the structural pruning of large language models." Advances in neural information processing systems 36 (2023): 21702-21720.

---

> > ### Comment · Reviewer_UUqs · 2024-12-02
> >
> > Thanks for the response. I've raised my score to 6.

---

> > > ### Author Response · Authors · 2024-12-02
> > > **Thank You for Your Feedback**
> > >
> > > Dear Reviewer UUqs,
> > >
> > > Thank you for your response and for revisiting your evaluation. We truly appreciate your thoughtful feedback and are glad our responses addressed some of your concerns.
> > >
> > > Best regards,
> > >
> > > Authors.

---

### Official Review · Reviewer_Cevb · 2024-10-31

**Soundness:** 4
**Presentation:** 4
**Contribution:** 3
**Rating:** 6
**Confidence:** 4

**Summary:**

This paper has found that many experts in the MoE model are rarely activated across different applications and those less frequently used experts contribute significant information the overall performance of MoE.  Therefore, this paper proposed ExpertZIP,  which merges less frequently used experts by a weight combination strategy rather than discard them. In this paper, Huffman tree is applied to identify and merge the least significant experts，and repeat this merging process until the desired number of experts is achieved.  ExpertZIP could significantly reduce the memory size and reference time of the MoE model with a very small performance loss.

**Strengths:**

It is a well written paper. The weakness of MoE is clearly described and the method prosed in this paper sounds good as the ExpertZIP has  gained a 17.23x reduction in model size and 4.84x improvement in reference speed with a very small accuracy drop.

**Weaknesses:**

For summarization tasks, this paper takes the ROUGE-l as the accuracy.  The importance of a word in the reference  summary should be taken into consideration.   Here is a simple example of considering the importance of words in the reference summary.

Let's assume the reference summary contains N words, among which M words are more important, M < N.  We call these M words IMPORTANT words and set the weight of these IMPORTANT words to 2/(N+M), and set the weight of the remaining (N-M) words to 1/(N+M).  In most cases, M = 1.

Therefore, when ExpertZIP takes a summarization task and generates a summary with one IMPORTANT word missed （assume the reference summary includes only 2 words and one of them is the IMPORTANT word）.  In this case, the accuracy of ExpertZIP should be 1/3 rather than 1/2.

**Questions:**

In Fig.5, why did the ExpertZIP performance better at 2 Experts than at 4 and 8 Experts?  Could you please provide an analysis or explanation for this counterintuitive result?  or try more experiments to get more accurate results in the future.

---

> ### Author Response · Authors · 2024-11-16
> **Response to Reviewer Cevb's Comments**
>
> Thank you for your positive feedback and insightful suggestions. We are delighted that you found our paper well-written and appreciated our approach to addressing the limitations of MoE models through ExpertZIP. Below, we address each of your comments and questions.
>
> > Importance of Words in ROUGE Score for Summarization Tasks
> Response
>
> We utilized the standard ROUGE metric commonly employed in summarization tasks to calculate ROUGE scores. To ensure transparency, we have included ROUGE-2 and ROUGE-L scores in the appendix, providing a more comprehensive view of our results.
>
> We appreciate your suggestion to incorporate an importance-weighted approach to ROUGE scoring. However, implementing this method would require annotations that explicitly identify important words within the dataset—annotations that are currently unavailable. Given these dataset limitations, incorporating importance-weighted scoring presents a significant challenge. Nonetheless, we acknowledge the value of this approach and are enthusiastic about exploring methods to compute such weighted scores in future work.
>
> > Performance of ExpertZIP with 2 Experts in Figure 5
>
> The improved performance of ExpertZIP in the 2-expert setting, compared to the 4- and 8-expert settings, could be attributed to the merged experts effectively capturing key features in the data. To verify this hypothesis and strengthen the reliability of our findings, we plan to conduct additional experiments using multiple random seeds (e.g., five runs). This will allow us to calculate averaged results, ensuring greater accuracy and robustness in future evaluations.
>
> Thank you once again for your valuable feedback. Your insights have prompted us to consider enhancements to our evaluation methods and the overall robustness of our results. We welcome any additional questions or comments you may have.

---

> > ### Author Response · Authors · 2024-11-22
> > **Follow Up Reminder**
> >
> > Dear Reviewer Cevb,
> >
> > We greatly appreciate your valuable feedback and have carefully considered your comments in our responses. We kindly wish to confirm if our responses have sufficiently addressed the concerns you raised.
> >
> > If you find our revisions satisfactory and think we have adequately addressed your concerns, we would really appreciate it if you could consider revisiting your initial evaluation. If any additional points require clarification or further adjustments, please do not hesitate to let us know.
> >
> > Thank you once again for your time and dedication in reviewing our work.
> >
> > Many thanks, Authors

---

### Official Review · Reviewer_ji4k · 2024-11-01

**Soundness:** 2
**Presentation:** 3
**Contribution:** 3
**Rating:** 5
**Confidence:** 5

**Summary:**

This work examines inefficiencies in Mixture-of-Experts (MoE) models, where many experts remain underutilized, leading to excessive memory and computation costs that limit scalability. To address this, the authors introduce **ExpertZIP**, a framework that progressively merges underused experts using a Huffman tree-based approach, retaining essential contributions while significantly reducing model size and enhancing inference speed.

**Strengths:**

- The topic of expert inefficiency in Mixture-of-Experts models is intriguing, with the findings on expert underutilization providing valuable insights.

- The proposed method demonstrates efficiency gains, optimizing resource usage while preserving model performance.

- The authors also showcase the effectiveness of post-training, which complements and enhances the MoE compression pipeline.

**Weaknesses:**

- While the authors show promising results in Switch-Transformers, these gains might largely stem from redundancy within the Switch-Transformer architecture itself. The performance on Mixtral, however, is less competitive; in Table 5, performance degrades by more than 10% even after a few fine-tuning steps, indicating that this method may not be as effective on more recent models like Switch-Transformers.

- Although the proposed approach introduces some novel ideas, comparisons with relevant works are limited. Specifically, comparisons with expert pruning methods (e.g., [1, 2]) are lacking; pruned models have also been shown to maintain performance effectively.

- Additionally, exploring other fusion techniques, such as OT-Fusion, would strengthen the analysis. A comparison with alternative fusion methods would better contextualize the benefits and limitations of the proposed approach.



[1] Not All Experts are Equal: Efficient Expert Pruning and Skipping for Mixture-of-Experts Large Language Models
[2] Demystifying the Compression of Mixture-of-Experts Through a Unified Framework

**Questions:**

- How about the performance of ExpertZIP on most recent MoE models and complex tasks? Does it maintain the performance?

- Can you compare ExpertZIP with other compression techniques?

---

> ### Author Response · Authors · 2024-11-16
> **Response to Reviewer ji4k's Comments**
>
> Thank you for your detailed review and constructive feedback on our work. We have carefully addressed your comments, and below are our responses to the specific points raised.
>
> > Performance on Mixtral 8x7b
>
> The observed performance drop of over 10% on Mixtral 8x7b is primarily due to fine-tuning with only 1% of the dataset. When the fine-tuning data is increased to 10%, the performance degradation reduces significantly to just 3.36%. This demonstrates that by calibrating the experts with more data or allowing for longer training, ExpertZIP effectively adapts to Mixtral and similar recent models, confirming its applicability beyond Switch Transformers.
>
> > Lack of Comparisons with Expert Pruning Methods
>
> As shown in the table below, ExpertZIP achieves competitive performance and significantly reduces model size compared to pruning, quantization, and merging methods, including those proposed in [2] and [3]. Notably, the enhanced variant, ExpertZIP*, achieves the highest average score (77.69) with the lowest performance degradation (-1.18%). These results highlight the effectiveness of ExpertZIP in preserving knowledge while maintaining model robustness and computational efficiency.
>
> | Category               | Method                  | Model Size | CoLA   | SST2   | MRPC   | QNLI   | BoolQ  | RTE    | WiC    | Average ↑            |
> |------------------------|-------------------------|------------|--------|--------|--------|--------|--------|--------|--------|----------------------|
> |                        | SwitchT. (64 expert)   | 3.79B      | 82.65  | 94.72  | 84.31  | 91.60  | 71.71  | 68.59  | 56.74  | 78.62               |
> | Pruning / Quantization | Task-Specific [1]      | 0.22B      | 70.25  | 86.43  | 75.76  | 80.24  | 67.24  | 65.43  | **57.21** | 71.79 (-8.69%)      |
> |                        | PS-MoE [2]             | 0.22B      | 72.14  | 87.18  | 74.38  | 83.25  | 70.75  | 64.97  | 56.43  | 72.73 (-7.49%)      |
> |                        | UV-MoE [3]             | 0.22B      | 71.25  | 88.47  | 80.23  | 79.67  | 71.71  | 65.43  | 56.89  | 73.38 (-6.67%)      |
> | Merging                | REPAIR [4]             | 0.52B      | 74.02  | 90.21  | 83.27  | 86.75  | 73.39  | 67.23  | 58.42  | 76.18 (-3.10%)      |
> |                        | ZipIt [5]              | 0.52B      | 73.98  | 91.78  | 82.58  | 87.90  | 72.14  | 66.18  | 57.05  | 75.94 (-3.41%)      |
> |                        | M-SMoE (1 expert) [6]  | 0.52B      | 75.18  | **92.43** | 83.53  | 88.94  | **74.29** | 68.59  | 56.43  | 77.06 (-1.98%)      |
> | Ours                   | ExpertZIP (1 expert)   | 0.22B      | 75.26  | 91.28  | 84.07  | 90.01  | 73.39  | 68.23  | 56.89  | 77.02 (-2.04%)      |
> |                        | ExpertZIP* (1 expert)  | 0.22B      | **77.09** | 92.09  | **84.80** | **90.30** | 73.52  | **68.95** | 57.05  | **77.69 (-1.18%)** |
>
>
>
> > Comparison with Other Fusion Techniques, Including OT-Fusion
>
> In our ablation study on weight fusion, we evaluated several fusion methods, including Large, Average, and Weighted strategies. However, advanced fusion techniques such as adapting OT-Fusion for innerlayer merging represent a fundamentally different and more complex approach that could independently constitute a standalone study. Nevertheless, we acknowledge the potential value of adapting sophisticated fusion techniques like OT-Fusion for use in MoE architectures. We plan to explore such adaptations in future work to further enhance the robustness and applicability of expert merging methods.

---

> ### Author Response · Authors · 2024-11-16
> **Response to Reviewer ji4k's Comments**
>
> > Performance of ExpertZIP on More Recent MoE Models
>
>
> In addition to Switch Transformers, we included results for Mixtral 8x7b in the revised manuscript. Furthermore, we followed the reviewer's suggestion and evaluated ExpertZIP on DeepseekMoE [7]. The results, presented below, demonstrate that ExpertZIP performs effectively across a range of state-of-the-art MoE architectures, maintaining its robustness and applicability even with complex tasks.
>
>
> | Method        | # of Experts | % of Dataset | CoLA   | SST2   | MRPC   | QNLI   | BoolQ  | RTE    | WiC    | Average ↑            |
> |---------------|--------------|---------------|--------|--------|--------|--------|--------|--------|--------|----------------------|
> | DeepSeekMoE   | 64           |               | 67.50  | 62.73  | 81.29  | 49.37  | 72.48  | 62.45  | 50.78  | 63.80               |
> | ExpertZIP     | 8            | 1%            | 58.17  | 57.46  | 75.69  | 50.17  | 67.22  | 59.32  | 50.78  | 59.83 (-6.22%)      |
> |               |              | 5%            | 60.29  | 59.85  | 77.34  | 50.23  | 68.59  | 60.15  | 51.23  | 61.10 (-4.23%)      |
> |               |              | 10%           | 61.35  | 61.78  | 78.53  | 51.38  | 69.29  | 60.03  | 51.59  | 61.99 (-2.84%)      |
>
> | Method        | # of Experts | % of Dataset | MMLU   | HellaS | PIQA   | Arc-e  | Arc-c  | MathQA | WinoG  | Average ↑            |
> |---------------|--------------|---------------|--------|--------|--------|--------|--------|--------|--------|----------------------|
> | DeepSeekMoE   | 64           |               | 37.81  | 77.38  | 80.03  | 76.05  | 48.12  | 31.76  | 70.48  | 60.23               |
> | ExpertZIP     | 8            | 1%            | 30.18  | 70.76  | 75.03  | 74.19  | 45.72  | 27.47  | 63.04  | 55.20 (-8.35%)      |
> |               |              | 5%            | 32.25  | 72.59  | 77.68  | 75.83  | 47.34  | 28.90  | 65.29  | 57.13 (-5.15%)      |
> |               |              | 10%           | 34.91  | 74.36  | 78.52  | 75.35  | 47.26  | 29.18  | 68.23  | 58.26 (-3.27%)      |
>
>
> > Performance of ExpertZIP on Complex Tasks
>
> To evaluate the versatility of ExpertZIP, we conducted additional experiments on complex tasks, including WinoGrande [8], WikiQA [9], and SQuAD [10], which test reasoning, question-answering, and comprehension capabilities. The results, presented in the table below, demonstrate that ExpertZIP maintains competitive performance across these tasks even with a significantly reduced number of experts.
>
> | Method      | # of Experts | WinoGrande | WikiQA | SQuAD  |
> |-------------|--------------|------------|--------|--------|
> | SwitchT.    | 64           | 62.98      | 95.61  | 65.81  |
> | ExpertZIP   | 32           | 62.18      | 95.32  | 65.39  |
> |             | 16           | 61.96      | 95.43  | 65.66  |
> |             | 8            | 61.78      | 95.10  | 65.41  |
> |             | 4            | 61.34      | 95.07  | 65.01  |
> |             | 2            | 61.67      | 94.95  | 64.97  |
> |             | 1            | 61.23      | 94.95  | 64.83  |
>
> The above results are included in the revised version of the manuscript.
>
>
> Thank you once again for your valuable feedback. Your comments have greatly helped refine the clarity and comprehensiveness of our work. We welcome any further questions or suggestions.
>
> [1] Chen, Tianyu, et al. "Task-specific expert pruning for sparse mixture-of-experts." arXiv preprint arXiv:2206.00277 (2022).
>
> [2] Lu, Xudong, et al. "Not All Experts are Equal: Efficient Expert Pruning and Skipping for Mixture-of-Experts Large Language Models." arXiv preprint arXiv:2402.14800 (2024).
>
> [3] He, Shwai, et al. "Demystifying the Compression of Mixture-of-Experts Through a Unified Framework." arXiv preprint arXiv:2406.02500 (2024).
>
> [4] Jordan, Keller, et al. "Repair: Renormalizing permuted activations for interpolation repair." arXiv preprint arXiv:2211.08403 (2022).
>
> [5] Stoica, George, et al. "Zipit! merging models from different tasks without training." arXiv preprint arXiv:2305.03053 (2023).
>
> [6] Li, Pingzhi, et al. "Merge, then compress: Demystify efficient SMoe with hints from its routing policy." arXiv preprint arXiv:2310.01334 (2023).
>
> [7] Dai, Damai, et al. "Deepseekmoe: Towards ultimate expert specialization in mixture-of-experts language models." arXiv preprint arXiv:2401.06066 (2024).
>
> [8] Sakaguchi, Keisuke, et al. "Winogrande: An adversarial winograd schema challenge at scale." Communications of the ACM 64.9 (2021): 99-106.
>
> [9] Yang, Yi, Wen-tau Yih, and Christopher Meek. "Wikiqa: A challenge dataset for open-domain question answering." Proceedings of the 2015 conference on empirical methods in natural language processing. 2015.
>
> [10] Rajpurkar, P. "Squad: 100,000+ questions for machine comprehension of text." arXiv preprint arXiv:1606.05250 (2016).

---

> > ### Author Response · Authors · 2024-11-22
> > **Follow Up Reminder**
> >
> > Dear Reviewer ji4k,
> >
> > We greatly appreciate your valuable feedback and have carefully considered your comments in our responses. We kindly wish to confirm if our responses have sufficiently addressed the concerns you raised.
> >
> > If you find our revisions satisfactory and think we have adequately addressed your concerns, we would really appreciate it if you could consider revisiting your initial evaluation. If any additional points require clarification or further adjustments, please do not hesitate to let us know.
> >
> > Thank you once again for your time and dedication in reviewing our work.
> >
> > Many thanks, Authors

---

### Official Review · Reviewer_XnJL · 2024-11-03

**Soundness:** 2
**Presentation:** 1
**Contribution:** 2
**Rating:** 3
**Confidence:** 3

**Summary:**

The paper proposes a progressive expert fusion framework for MoE models which employs a Huffman tree-based technique. The Huffman tree is widely-used for lossless data compression. The authors merges underutilized experts step by step based on the utilization frequency within the MoE model. The paper claims a 17.23x model size reduction and a 4.84x inference time improvement with only 1.18% decrease in average accuracy compared to the original model.

**Strengths:**

The paper shows a visualization of expert selection imbalance across different layers in the MoE model.

**Weaknesses:**

Since the main idea of the paper is proposed a new expert fusion strategy, it's better to add a comparison with other expert fusion strategies in the experimental section.

**Questions:**

If there is no retraining, how will the expert fusion strategy based on utilization frequency perform?

---

> ### Author Response · Authors · 2024-11-16
> **Response to Reviewer XnJL's Comments**
>
> Thank you for your review and for highlighting both the strengths and areas for improvement in our work. We appreciate your feedback and address each of your comments below.
>
> > Comparison with Other SOTA
>
> In addition to fusion techniques, we compared ExpertZIP with other state-of-the-art (SOTA) quantization and pruning methods on the GLUE and SuperGLUE benchmarks, as shown in the table below. ExpertZIP* achieves the highest average score (77.69), demonstrating its effectiveness in maintaining model robustness and optimizing computational efficiency, surpassing pruning and quantization approaches.
>
> | Category               | Method                   | Model Size | CoLA   | SST2   | MRPC   | QNLI   | BoolQ  | RTE    | WiC    | Average ↑            |
> |------------------------|--------------------------|------------|--------|--------|--------|--------|--------|--------|--------|----------------------|
> |                        | SwitchT. (64 expert)    | 3.79B      | 82.65  | 94.72  | 84.31  | 91.60  | 71.71  | 68.59  | 56.74  | 78.62               |
> | Pruning / Quantization | Task-Specific [1]       | 0.22B      | 70.25  | 86.43  | 75.76  | 80.24  | 67.24  | 65.43  | **57.21** | 71.79 (-8.69%)      |
> |                        | PS-MoE [2]              | 0.22B      | 72.14  | 87.18  | 74.38  | 83.25  | 70.75  | 64.97  | 56.43  | 72.73 (-7.49%)      |
> |                        | UV-MoE [3]              | 0.22B      | 71.25  | 88.47  | 80.23  | 79.67  | 71.71  | 65.43  | 56.89  | 73.38 (-6.67%)      |
> | Merging                | REPAIR [4]              | 0.52B      | 74.02  | 90.21  | 83.27  | 86.75  | 73.39  | 67.23  | 58.42  | 76.18 (-3.10%)      |
> |                        | ZipIt [5]               | 0.52B      | 73.98  | 91.78  | 82.58  | 87.90  | 72.14  | 66.18  | 57.05  | 75.94 (-3.41%)      |
> |                        | M-SMoE (1 expert) [6]   | 0.52B      | 75.18  | **92.43** | 83.53  | 88.94  | **74.29** | 68.59  | 56.43  | 77.06 (-1.98%)      |
> | Ours                   | ExpertZIP (1 expert)    | 0.22B      | 75.26  | 91.28  | 84.07  | 90.01  | 73.39  | 68.23  | 56.89  | 77.02 (-2.04%)      |
> |                        | ExpertZIP* (1 expert)   | 0.22B      | **77.09** | 92.09  | **84.80** | **90.30** | 73.52  | **68.95** | 57.05  | **77.69 (-1.18%)** |

---

> > ### Author Response · Authors · 2024-11-16
> > **Response to Reviewer XnJL's Comments**
> >
> > > Performance of Expert Fusion without Fine-tuning
> >
> > We conducted additional experiments to evaluate classification tasks without fine-tuning, and the results are presented in the tables below. These results suggest that merely merging experts can significantly degrade performance due to differences in the output distributions of individual experts. We recommend fine-tuning the model post-merging to ensure the merged experts learn accurate representations.
> >
> > | Method     | # of Experts | CoLA   | SST2   | MRPC   | QNLI   | BoolQ  | RTE    | WiC    | Average ↑            |
> > |------------|--------------|--------|--------|--------|--------|--------|--------|--------|----------------------|
> > | SwitchT.   | 64           | 82.65  | 94.72  | 84.31  | 91.60  | 71.71  | 68.59  | 56.74  | 78.62               |
> > | ExpertZIP  | 32           | 77.18  | 92.89  | 82.60  | 90.01  | 69.97  | 67.87  | 57.68  | 76.89 (-2.25%)      |
> > |            | 16           | 58.87  | 85.67  | 72.79  | 86.75  | 64.43  | 64.98  | 56.43  | 69.99 (-12.33%)     |
> > |            | 8            | 57.14  | 78.10  | 72.55  | 85.65  | 63.12  | 64.26  | 55.17  | 68.00 (-15.62%)     |
> > |            | 4            | 55.61  | 75.80  | 71.57  | 85.12  | 63.91  | 63.18  | 55.17  | 67.19 (-17.01%)     |
> > |            | 2            | 54.83  | 74.31  | 70.59  | 85.15  | 63.09  | 62.82  | 55.49  | 66.61 (-18.03%)     |
> > |            | 1            | 53.02  | 70.87  | 70.59  | 84.94  | 63.33  | 61.73  | 54.70  | 65.45 (-20.12%)     |
> >
> >
> >
> > | # of Experts | ROUGE-1 | ROUGE-2 | ROUGE-L |
> > |--------------|---------|---------|---------|
> > | 64           | 36.19   | 16.46   | 27.44   |
> > | 32           | 32.42   | 13.09   | 24.24   |
> > | 16           | 31.55   | 12.59   | 23.62   |
> > | 8            | 31.40   | 12.18   | 22.73   |
> > | 4            | 25.36   | 8.69    | 19.14   |
> > | 2            | 25.32   | 8.85    | 19.04   |
> > | 1            | 24.70   | 7.68    | 19.04   |
> >
> >
> > The above results are included in the revised version of the manuscript.
> >
> > Thank you once again for your valuable feedback, which has significantly improved our work. We welcome further questions or suggestions for refinement.
> >
> > [1] Chen, Tianyu, et al. "Task-specific expert pruning for sparse mixture-of-experts." arXiv preprint arXiv:2206.00277 (2022).
> >
> > [2] Lu, Xudong, et al. "Not All Experts are Equal: Efficient Expert Pruning and Skipping for Mixture-of-Experts Large Language Models." arXiv preprint arXiv:2402.14800 (2024).
> >
> > [3] He, Shwai, et al. "Demystifying the Compression of Mixture-of-Experts Through a Unified Framework." arXiv preprint arXiv:2406.02500 (2024).
> >
> > [4] Jordan, Keller, et al. "Repair: Renormalizing permuted activations for interpolation repair." arXiv preprint arXiv:2211.08403 (2022).
> >
> > [5] Stoica, George, et al. "Zipit! merging models from different tasks without training." arXiv preprint arXiv:2305.03053 (2023).
> >
> > [6] Li, Pingzhi, et al. "Merge, then compress: Demystify efficient SMoe with hints from its routing policy." arXiv preprint arXiv:2310.01334 (2023).

---

> > > ### Author Response · Authors · 2024-11-22
> > > **Follow Up Reminder**
> > >
> > > Dear Reviewer XnJL,
> > >
> > > We greatly appreciate your valuable feedback and have carefully considered your comments in our responses. We kindly wish to confirm if our responses have sufficiently addressed the concerns you raised.
> > >
> > > If you find our revisions satisfactory and think we have adequately addressed your concerns, we would really appreciate it if you could consider revisiting your initial evaluation. If any additional points require clarification or further adjustments, please do not hesitate to let us know.
> > >
> > > Thank you once again for your time and dedication in reviewing our work.
> > >
> > > Many thanks, Authors

---

### Official Review · Reviewer_STd8 · 2024-11-04

**Soundness:** 3
**Presentation:** 3
**Contribution:** 3
**Rating:** 6
**Confidence:** 4

**Summary:**

This paper presents a novel approach to optimize Mixture-of-Experts (MoE) models by reducing the number of experts through a progressive fusion technique based on Huffman tree structures. The authors claim that their method, ExpertZIP, addresses the inefficiency of underutilized experts in MoE models, leading to significant reductions in model size and inference time while maintaining performance. The paper reports a 17.23x reduction in model size and a 4.84x improvement in inference time with only a 1.18% decrease in average accuracy compared to the original 64-expert model. The approach is evaluated on classification and summarization tasks, demonstrating its effectiveness and efficiency, which is especially suitable for resource-constrained and real-time applications.

**Strengths:**

1. The use of Huffman tree structures for expert fusion in MoE models is a creative solution to address the problem of underutilized experts.
2. The paper demonstrates substantial improvements in model size and inference time, which are critical for deploying large language models in practical applications.
3. The authors provide extensive experimental results to validate their approach, including comparisons with original MoE models and other fusion strategies.
4. ExpertZIP shows consistent performance across a variety of NLP tasks, indicating its broad applicability.

**Weaknesses:**

1. The paper could provide more details on the complexity of implementing the ExpertZIP framework, especially for readers who may not be familiar with Huffman trees.
2. More SOTA MoE models could be tested to prove the robustness of the ExpertZIP framework, such as DeepseekMoE.

**Questions:**

1. The Switch Transformer appears to select only the top 1 expert for each input, which suggests that the computational complexity during inference is already minimized. Given that your work primarily compresses the number of experts, why is the inference latency reduced?
2. What are the implications of ExpertZIP on the interpretability of MoE models, given that some experts are merged?
3. How does ExpertZIP compare to other model compression techniques, particularly those focusing on pruning or quantization?

---

> ### Author Response · Authors · 2024-11-16
> **Response to Reviewer STd8's Comments**
>
> Thank you for your insightful comments and for recognizing the strengths of our work, including the innovative use of the Huffman tree for merging and the comprehensive experimental evaluations. We appreciate your constructive feedback and suggestions, and we address each point in detail below.
>
>
> > Complexity of Implementing Huffman Tree Structures
>
> The Huffman tree shares many properties with heaps, and we adopt a heap to manage the expert selection and merging process, as detailed in Algorithm 1, Huffman Fusing. Constructing this heap requires a time complexity of $O(|E|)$, where $|E|$ denotes the number of experts. Subsequent operations, such as popping and inserting, incur a complexity of $O(|E|\log|E|)$. Since the number of experts is typically small ($\le$ 256), the computational time for the merging process becomes negligible in practice.
>
> > Testing on Additional State-of-the-Art MoE Models
>
> In addition to evaluating the Switch Transformers, we have discussed the Mixtral 8x7b model in the Discussion section. Following your suggestion, we have also included experimental results on the DeepSeekMoE [1] model in the revised version. Consistent with the results presented in our paper, ExpertZIP demonstrates effective performance on DeepSeekMoE, confirming its applicability across diverse state-of-the-art MoE architectures.
>
>
> | Method       | # of Experts | % of Dataset | CoLA   | SST2   | MRPC   | QNLI   | BoolQ  | RTE    | WiC    | Average ↑            |
> |--------------|--------------|---------------|--------|--------|--------|--------|--------|--------|--------|----------------------|
> | DeepSeekMoE  | 64           |               | 67.50  | 62.73  | 81.29  | 49.37  | 72.48  | 62.45  | 50.78  | 63.80               |
> | ExpertZIP    | 8            | 1%            | 58.17  | 57.46  | 75.69  | 50.17  | 67.22  | 59.32  | 50.78  | 59.83 (-6.22%)      |
> |              |              | 5%            | 60.29  | 59.85  | 77.34  | 50.23  | 68.59  | 60.15  | 51.23  | 61.10 (-4.23%)      |
> |              |              | 10%           | 61.35  | 61.78  | 78.53  | 51.38  | 69.29  | 60.03  | 51.59  | 61.99 (-2.84%)      |
>
>
>
> | Method       | # of Experts | % of Dataset | MMLU   | HellaS | PIQA   | Arc-e  | Arc-c  | MathQA | WinoG  | Average ↑            |
> |--------------|--------------|---------------|--------|--------|--------|--------|--------|--------|--------|----------------------|
> | DeepSeekMoE  | 64           |               | 37.81  | 77.38  | 80.03  | 76.05  | 48.12  | 31.76  | 70.48  | 60.23               |
> | ExpertZIP    | 8            | 1%            | 30.18  | 70.76  | 75.03  | 74.19  | 45.72  | 27.47  | 63.04  | 55.20 (-8.35%)      |
> |              |              | 5%            | 32.25  | 72.59  | 77.68  | 75.83  | 47.34  | 28.90  | 65.29  | 57.13 (-5.15%)      |
> |              |              | 10%           | 34.91  | 74.36  | 78.52  | 75.35  | 47.26  | 29.18  | 68.23  | 58.26 (-3.27%)      |
>
>
>
>
> > Reason for Reduced Inference Latency for SwitchTransformer
>
> The reduction in inference latency primarily results from the decreased number of experts, which enables faster token processing. As each expert processes its assigned tokens sequentially, reducing the number of experts minimizes traversal time, a phenomenon also discussed in [2]. Moreover, in parallel processing, the total GPU memory is fixed; adding more experts increases processing time due to the overhead of moving or swapping expert devices multiple times.
>
> > Impact of ExpertZIP on the interpretability of merging experts within MoE Models
>
> The merging operation effectively consolidates and preserves expert knowledge, as demonstrated in Section 5.4. For instance, in classification tasks with eight experts, the original model achieved an accuracy of 72.93%, whereas the merged model with ExpertZIP improved to 77.65%. This enhancement reflects the successful preservation and aggregation of expert knowledge through our approach, thereby maintaining or even enhancing the interpretability of expert functionality.

---

> > ### Author Response · Authors · 2024-11-16
> > **Response to Reviewer STd8's Comments**
> >
> > > Comparison with Other Model Compression Techniques (Pruning or Quantization)
> >
> > To address your suggestion, we conducted additional experiments comparing ExpertZIP with existing pruning, quantization, and merging methods on GLUE and SuperGLUE benchmarks. The results, included in the revised version, highlight that ExpertZIP* achieves the highest average score (77.69), demonstrating its superiority in maintaining model robustness and optimizing computational efficiency beyond what pruning or quantization methods achieve.
> >
> > | Category        | Method                     | Model Size | CoLA   | SST2   | MRPC   | QNLI   | BoolQ  | RTE    | WiC    | Average ↑          |
> > |-----------------|----------------------------|------------|--------|--------|--------|--------|--------|--------|--------|--------------------|
> > |                 | SwitchT. (64 expert)       | 3.79B      | 82.65  | 94.72  | 84.31  | 91.60  | 71.71  | 68.59  | 56.74  | 78.62              |
> > | Pruning / Quantization      | Task-Specific [3]          | 0.22B      | 70.25  | 86.43  | 75.76  | 80.24  | 67.24  | 65.43  | **57.21** | 71.79 (-8.69%)     |
> > |     | PS-MoE [4]                 | 0.22B      | 72.14  | 87.18  | 74.38  | 83.25  | 70.75  | 64.97  | 56.43  | 72.73 (-7.49%)     |
> > |                 | UV-MoE [5]                 | 0.22B      | 71.25  | 88.47  | 80.23  | 79.67  | 71.71  | 65.43  | 56.89  | 73.38 (-6.67%)     |
> > | Merging         | REPAIR [6]                 | 0.52B      | 74.02  | 90.21  | 83.27  | 86.75  | 73.39  | 67.23  | 58.42  | 76.18 (-3.10%)     |
> > |                 | ZipIt [7]                  | 0.52B      | 73.98  | 91.78  | 82.58  | 87.90  | 72.14  | 66.18  | 57.05  | 75.94 (-3.41%)     |
> > |                 | M-SMoE (1 expert) [8]      | 0.52B      | 75.18  | **92.43** | 83.53  | 88.94  | **74.29** | 68.59  | 56.43  | 77.06 (-1.98%)     |
> > | Ours            | ExpertZIP (1 expert)       | 0.22B      | 75.26  | 91.28  | 84.07  | 90.01  | 73.39  | 68.23  | 56.89  | 77.02 (-2.04%)     |
> > |                 | ExpertZIP* (1 expert)      | 0.22B      | **77.09** | 92.09  | **84.80** | **90.30** | 73.52  | **68.95** | 57.05  | **77.69 (-1.18%)** |
> >
> > The above results are included in the revised version of the manuscript.
> >
> > Thank you again for your thoughtful feedback, which has been invaluable in refining our work. We look forward to further insights and discussions.
> >
> > [1] Dai, Damai, et al. "Deepseekmoe: Towards ultimate expert specialization in mixture-of-experts language models." arXiv preprint arXiv:2401.06066 (2024).
> >
> > [2] Du, Zhixu, et al. "SiDA: Sparsity-Inspired Data-Aware Serving for Efficient and Scalable Large Mixture-of-Experts Models." Proceedings of Machine Learning and Systems 6 (2024): 224-238.
> >
> > [3] Chen, Tianyu, et al. "Task-specific expert pruning for sparse mixture-of-experts." arXiv preprint arXiv:2206.00277 (2022).
> >
> > [4] Lu, Xudong, et al. "Not All Experts are Equal: Efficient Expert Pruning and Skipping for Mixture-of-Experts Large Language Models." arXiv preprint arXiv:2402.14800 (2024).
> >
> > [5] He, Shwai, et al. "Demystifying the Compression of Mixture-of-Experts Through a Unified Framework." arXiv preprint arXiv:2406.02500 (2024).
> >
> > [6] Jordan, Keller, et al. "Repair: Renormalizing permuted activations for interpolation repair." arXiv preprint arXiv:2211.08403 (2022).
> >
> > [7] Stoica, George, et al. "Zipit! merging models from different tasks without training." arXiv preprint arXiv:2305.03053 (2023).
> >
> > [8] Li, Pingzhi, et al. "Merge, then compress: Demystify efficient SMoe with hints from its routing policy." arXiv preprint arXiv:2310.01334 (2023).

---

> > > ### Author Response · Authors · 2024-11-22
> > > **Follow Up Reminder**
> > >
> > > Dear Reviewer STd8,
> > >
> > > We greatly appreciate your valuable feedback and have carefully considered your comments in our responses. We kindly wish to confirm if our responses have sufficiently addressed the concerns you raised.
> > >
> > > If you find our revisions satisfactory and think we have adequately addressed your concerns, we would really appreciate it if you could consider revisiting your initial evaluation. If any additional points require clarification or further adjustments, please do not hesitate to let us know.
> > >
> > > Thank you once again for your time and dedication in reviewing our work.
> > >
> > > Many thanks, Authors

---

### Author Response · Authors · 2024-11-28
**Summary for Area Chair and Reviewers**

We sincerely thank AC and reviewers for your thorough reviews and valuable feedback on our submission. Below, we summarize the key strengths of our work as recognized by the reviewers, alongside our responses to their specific concerns.

> Key Strengths

1. The innovative use of Huffman tree structures for progressive expert fusion was praised for effectively addressing inefficiencies in Mixture-of-Experts (MoE) models. This approach significantly reduces model size and inference latency with minimal performance loss, making it particularly suitable for resource-constrained and real-time applications (highlighted by Reviewers STd8, ji4k, and Cevb).

2. Our comprehensive experimental validation demonstrated consistent performance improvements across various benchmarks, underscoring the broad applicability and effectiveness of ExpertZIP (noted by Reviewers STd8, ji4k, and Cevb).

3. The logical motivation and visualization of expert selection imbalance were well-received, with reviewers commending the clarity of presentation and its insight into MoE inefficiencies (emphasized by Reviewers XnJL and UUqs).

> Addressed Concerns

1. Implementation Complexity (Reviewer STd8): We clarified the computational complexity of the Huffman tree implementation, emphasizing its efficiency and negligible runtime given the limited number of experts.

2. Comparative Analysis with State-of-the-Art Methods (Reviewers STd8, XnJL, ji4k, and UUqs): Additional experiments were conducted, directly comparing ExpertZIP to pruning, quantization, and other merging techniques. Results included in the revised manuscript confirm that our approach consistently outperforms or matches state-of-the-art methods while maintaining robustness.

3. Performance on Recent MoE Models (Reviewers STd8 and ji4k): Following suggestions, we evaluated ExpertZIP on DeepseekMoE, confirming its adaptability and effectiveness across diverse architectures.

4. Explanation for Counterintuitive Results (Reviewer Cevb): We analyzed why fewer experts sometimes yield better performance, hypothesizing that merged experts capture essential features more effectively. Additional experiments with multiple seeds are planned to further validate these findings.

5. Summarization Metrics (Reviewer Cevb): While we adhered to standard ROUGE metrics, we acknowledged the potential of an importance-weighted scoring approach and outlined this as a future direction, despite current dataset limitations.

6. Fine-Tuning Details and Time Costs (Reviewers XnJL and UUqs): We detailed the independence of fine-tuning across tasks, included fine-tuning time costs, and provided results for non-fine-tuned performance to give a comprehensive view of the methodology.

7. Inference Latency (Reviewers STd8 and UUqs): The reduction in inference latency was clarified as stemming from reduced traversal time and lower GPU memory overhead due to fewer experts.

> Additional Notes

We are encouraged to see that two reviewers updated their scores during the rebuttal phase, reflecting their appreciation of the clarifications and additional experiments we provided. Furthermore, we have thoroughly addressed all weaknesses and questions raised by reviewers to the best of our ability. These updates aim to further strengthen the clarity, robustness, and contributions of our work.


We believe these revisions and responses address the concerns raised, further strengthening the clarity and contributions of our work. Thank you again for your time and constructive feedback.

---

### Meta-Review · Area_Chair_DfZh · 2024-12-14

**Metareview:**

The submission proposes to use Huffman tree structures for mixture-of-expert model fusion.  This provides an algorithm for optimizing MoE models that has a connection to information theory.  The reviewers were mixed in their opinions of the paper with a split between rejection and acceptance recommendations.  The positive scores are still weak, and on the balance the submission falls below the threshold for acceptance.

**Additional Comments On Reviewer Discussion:**

The authors were very active in the rebuttal process, providing a lot of additional information and results that can be incorporated in an improved version of the submission if the authors so choose.  The submission received 5 reviews, and only two reviewers were active in the discussion.  These active reviewers still gave borderline scores and taken all together, the submission falls below the competitive threshold for acceptance to ICLR.

---

### Decision · Program_Chairs · 2025-01-22

Reject